# Flow-Shop Predictive Modeling for Multi-Automated Guided Vehicles Scheduling in Smart Spinning Cyber–Physical Production Systems

**Basit Farooq**, **Jinsong Bao** * **and Qingwen Ma**

College of Mechanical Engineering, Donghua University, Shanghai 201620, China;
basitfarooq@mail.dhu.edu.cn (B.F.); 15251811926@163.com (Q.M.)

* Correspondence: bao@dhu.edu.cn; Tel.: +0086-139-1635-1103

**Abstract:** Pointed at a problem that leads to the high complexity of the production management tasks in the multi-stage spinning industry, mixed flow batch production is often the case in response to a customer's personalized demands. Manual handling cans have a large number of tasks, and there is a long turnover period in their semi-finished products. A novel heuristic research was conducted that considered mixed-flow shop scheduling problems with automated guided vehicle (AGV) distribution and path planning to prevent conflict and deadlock by optimizing distribution efficiency and improving the automation degree of can distribution in a draw-out workshop. In this paper, a cross-region shared resource pool and an inter-regional independent resource pool, two AGV predictive scheduling strategies are established for the ring-spinning combing process. Besides completion time, AGV utilization rate and unit AGV time also analyzed with the bottleneck process of the production line. The results of the optimal computational experiment prove that a draw frame equipped with multi-AGV and coordinated scheduling optimization will significantly improve the efficiency of can distribution. Flow-shop predictive modeling for multi-AGV resources is scarce in the literature, even though this modeling also produces, for each AGV, a control mode and, if essential, a preventive maintenance plan.

**Keywords:** smart spinning; mixed flow-shop; multi-AGV predictive modeling; cyber–physical production system

---

## 1. Introduction

Spinning is the lifeblood industry of the textile industry, where the fiber is drawn-out, twisted, and wrapped onto a bobbin by using state-of-the-art twisting techniques. The ring-spinning production process includes opening and cleaning (opening, blending, and cleaning take place in a blow room), carding, pre-combed drawing, lap forming, combing, post-combed drawing, simplex, ring-spinning, and winding. The spinning process is dynamic, implicating multiple discrete workshops, and around more than 80 production pieces of equipment are used during it. The spindle's speed when spinning can reach up to 25,000 r/min and the spindle is the leading apparatus of these extremely sophisticated systems. Though these smart, or intelligent systems, are relatively efficient, they also suffer from various problems; particularly in the draw-out workshop, manual handling can distribution has a large number of demanding tasks, and there is a long turnover period in semi-finished products.

Recent years have seen an increase in the amount of the flow-shops with identical counterpart machines, like those in the spinning industry, that have gone through upgrades towards becoming intelligent automation industries from labor-intense sectors [1]. As discussed in [2], a smart spinning system is a system that contains multiple computing elements (sensors and actuators) and a processing unit, all managed by restraining the data. Based on flow of information, the data are pre-processed and a command is sent to the actuator to perform a pre-programmed action that can detect changes in their surroundings and react to them to produce a practical outcome. Scheduling is inevitable in these smart spinning cyber–physical production systems (CPPS). Due to their strong computational, analytical, and processing capabilities, intelligent mechatronic systems like automated guided vehicles (AGVs) are vital to assure stability and competence [3]. These smart transport systems equipped with radio waves, vision cameras, magnets, or lasers for navigation (automatic guiding devices) that can walk-along a preset guidance path to complete a series of horizontal transport operations. An approach developed in [4] at a particular changing state of moving vehicles and production equipment had the feature of processing multiple semi-finished products at the same time within the premises of production logistics. A distributed method of discrete-event simulation for highly distributed manufacturing systems was discussed in [5]. The required time to complete work orders, the amount of energy used to transport the loads, and the utilization of work stations are the highest performance standards for AGV material handling systems [6,7], all of which are widely used to relocate material in modern, flexible manufacturing systems [8] and which affect the effectiveness of manufacturing processes by enhancing their efficiency. Though the benefits of allowing for AGV job preemptions can be significant, few studies have considered pre-emptive scheduling in the context of flow-shop predictive modeling. For example, in [9,10], AGV dispatching and cooperative waterborne AGVs prediction models were addressed, and a class of precedence constraints was proposed.

Various literature reviews have been proposed during the last decade about the impact of AGVs in smart industries. AGVs have economic, environmental, and technical advantages. They also come with three main problems: AGV scheduling, AGV path planning, and AGV control system implementation [11–13]. AGVs scheduling of a more flexible, modular, and intelligent manufacturing system does not require much time and budget for initial installation. Stetter et al. developed a virtual diagnostic sensor design for production tasks to solve the problem of task scheduling and coordination control by an AGV system [14]. A concise overview of guide-path design, estimating the number of locations and parking, pickup, and delivery points that considered the number of required vehicles was given by Małopolski [15], and a fault control strategy was given by Witczak [16]. There have been fewer scholarly studies on the battery management of AGVs to obtain more productive hours and increase the flexibility of manufacturing systems—still, an effort was made by Kabir et al. in this regard [17]. A cooperative AGV-based production system depends on its efficiency, effectiveness, availability, and reliability by predicting AGV-battery remaining useful life (RUL) [18–20]. CPPS-based investigations and solutions, i.e., the bottleneck of supply chain management for material handling and production manufacturing, have been presented for complex material flows [21–24]. When answering this problem, one must cope with difficulties connected with flow-shop scheduling, as well as those imposed by the resource constraints that must be satisfied in the whole system at every moment [25–27].

This work was inspired to resolve the confronting problems of scheduling in real-time can distribution and path planning in the continuous production of spinning by using a mixed flow-shop predictive modeling method. The related problems are as follows: (a) How to solve a general problem of AGV scheduling and path planning to prevent conflict and deadlock in parallel machines flow-shop? How does a spinning CPPS deal with scheduling tasks in mass production? Additionally, how to deploy AGVs to work together to process real-time tasks?

The main contributions of this paper are as follows:

1. To effectively reduce the makespans and total completion time. To the best of our knowledge, this is the first time a novel approach that handles both cross-region shared resource and inter-regional independent resource pools.

2. Based on the intended categories of scheduling tasks, an AGV transportation route strategy is developed for mass production in spinning CPPS.

3. A mathematical model for real-time task processing for dissimilar cotton and polyester processing in multi-AGV scheduling is designed to prevent conflict and deadlock by assigning different tasks: AGV assignments, AGV sorting, and task sources.

Our results demonstrate the adequacy of the presented methods when the number of AGVs intensifies to a certain extent, as increasing the of number AGVs decreased the completion time drop sharply, which reduces the AGV utilization.

The paper is organized as follows: Section 2 imparts a comprehensive background of the analysis of the production process of the spinning workshop, a mathematical model of multi-AGV scheduling, model assumptions, object function, and the uniqueness constraint. Section 3 concentrates on the explanation of the multi-AGV scheduling simulation modeling, simulation model construction, and the AGV resource pool strategy-based bottleneck analysis. Section 4 emphasizes the results by doing a comparative study of two AGV resource pool strategies and a comprehensive analysis of multi-AGV scheduling in the process flow. Finally, conclusions are summarized in Section 5.

## 2. Analysis of the Production Process of the Spinning Workshop

We tried to examine previously-discussed drawing shops to improve our earlier presented approach in two key areas: First, the series structure, where the raw materials for each process only depend on one process and the same processed products used for another process, is shown in Figure 1. The second parallel structure, in which the raw material of each procedure depends on multiple processes or the product of one process used as the raw material of other various processes, is shown in Figure 2.

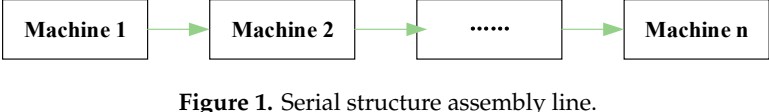

**Figure 1.** Serial structure assembly line.

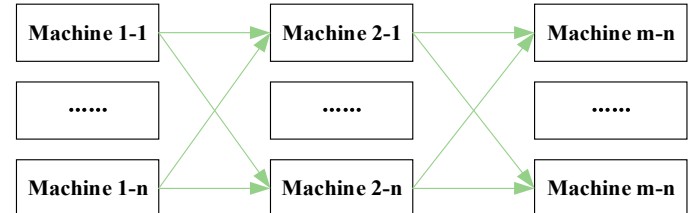

**Figure 2.** Parallel structure assembly line.

A traditional draw shop has a low-level of automation, and semi-finished products in various processes mainly rely on manual transportation. On the other hand, the efficiency of manual handling is poor, and it cannot meet the demand for rapid response, which has become a bottleneck for speeding up production lines, which causes higher labor costs and a more significant workload for employees, thus resulting in insufficient production. The usage of an AGV instead of manual handling between traditional production processes can effectively improve the automation level of a workshop [28–30]. By relying on information technology to enhance the accurate response capabilities of an AGV, it can achieve precisely guided tasks such as advanced stocking and rapid shipment, which avoid the traditional experience of employees [31,32]. The ability to rely on AGVs can effectively improve the production efficiency of a workshop.

A spinning workshop is a mixed-flow workshop. A typical production line contains multiple operational processes that involve various workshops. The corresponding process sequence must be satisfied with numerous methods. Therefore, a spinning workshop is regarded as a multi-stage parallel production assembly-line system. Figure 3 illustrates the ring-spinning production process of a spinning workshop, the first stage explains the opening and cleaning of the cotton bales and then the carding, mainly to remove impurities from cotton or polyester and to produce strip-shaped semi-finished products. The second stage is primarily to pre-draw and to draw after carding. The density of cotton or polyester is not uniform. Pre-drawing and drawing further mix the raw materials evenly, and mixing the two raw materials of cotton and polyester produces semi-finished products with different ratios. Phase 3 is thickness and thinness, and the purpose of it is further processing that makes the final products ready for the customer's demands according to the desired radius size. It can be observed from the figure that each stage in the flow-process includes multiple processes, each process performed by one machine or numerous identical machines, and each process relies on semi-finished products of one or more operations as raw materials. The mixing process in a spinning and drawing workshop uses cotton slivers produced in the combing process and polyester slivers produced in the polyester process as the raw materials. In contrast, the mixing process uses the products from the first process as raw materials. This article mainly studies the two-stage workshop in the production stage. For convenience, it is called the drawing-out flow-shop.

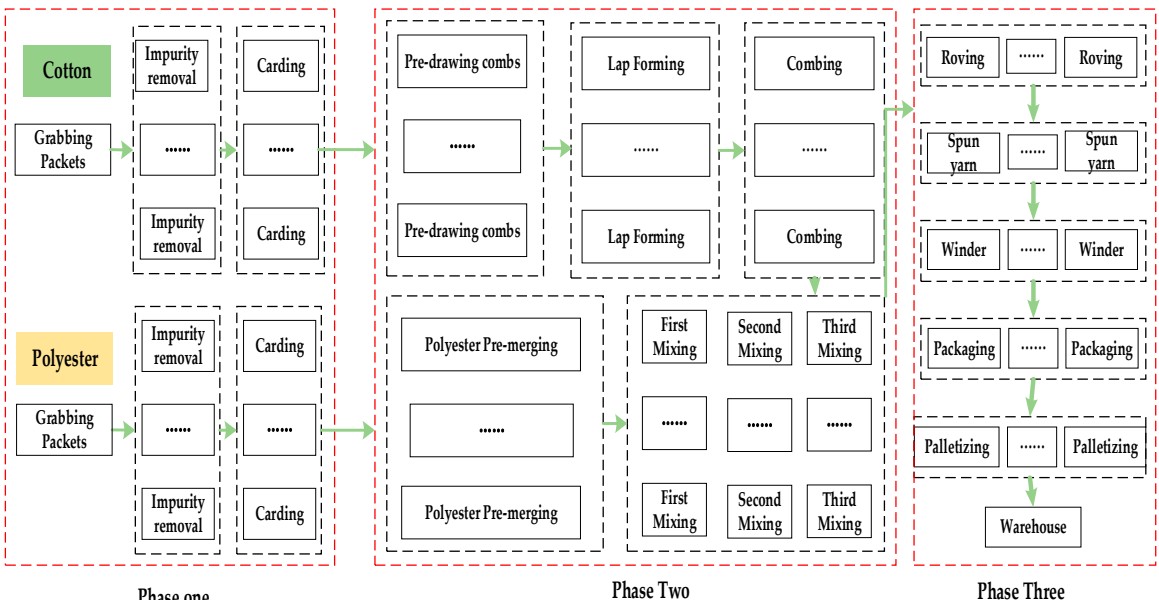

**Figure 3.** The production process of ring spinning in a spinning workshop.

### 2.1. Mathematical Model of Multi-AGV Scheduling

#### 2.1.1. Processing Equipment Definition

Multiple parallel processes at the same level represent multiple production lines of different semi-finished products, and a production line of semi-finished products is composed of multiple serial processes. Keep in mind that the workshop has *ith* level operations, $Z_i$ is the number of product lines under the *ith*-level operations, and $P_{iz}$ represents the number of continuing operations of the z-product line. Under each process, there are $K_{p_{iz}}$ identical pieces of processing equipment for processing, and $k_{p_{iz}}$ denotes the *kth* machine that executes the process $p_{iz}$. Each piece of processing equipment has different attributes: in addition to the production efficiency of the processing equipment, there is the production time per unit of product, the type and quantity of raw material required for processing, the batch produced the number of products processed in one batch, starting processing time, and processing

completion time. Among these, the production efficiency of the equipment, the type and quantity of raw materials, and the number of products obtained in one processing are the input of the model, and the other variables are the amounts of decisions.

### 2.1.2. Raw Material and Product Definition

Considering the particularity of assembly line production, the processing raw material of one device is a semi-finished product produced by another device, so the total number of products are represented by $j^+_{P_{iz}P_{i'z'}}$; $p'_{i'z'}$ is the process required for the process $p_{iz}$ processing, indicating the process $p_{iz}$ comes from the *Jth* raw material of the process; $p'_{i'z'}$ represents the total number of finished products processed by the process $p_{iz}$; and $J^-_{P_{iz}}$ represents the *Jth* finished product of the process $p_{iz}$.

In addition to the initial raw materials of the assembly line, all raw materials contain two attributes: one is the $p'_{i'z'}$ processing completion time of the process, and $p_{iz}$ is the transportation cut-off time of the process. The state of the material after the start of processing becomes the product status; therefore, one does not need to consider the initial processing properties of the material.

### 2.1.3. AGV Definition

Considering that there are $V$ identical AGVs in a spinning draw frame, each AGV transports between any two machines, and the transportation time depends on the actual physical distance and road conditions between the two machines.

An AGV car has two walking states in its scheduling system. One is without-load when the car receives a transportation task and it needs to travel from the current position to the starting position of the transport task without-load. The car is currently in the initial position of transportation without-load, and the distance covered by it is 0. The second is with-load, that is, the trolley runs from the initial position of the transportation task to the end position of the transportation task. Each AGV contains different attributes, including with-load start time, without-load end time, the without-load starting point for a transport task, load start time, load end time, load line for a transport task, and specific raw materials for means of transport.

### 2.1.4. Overall Variable Definition

Table 1 shows that the process $p_{iz}$ for each machine must start processing after collecting the number of raw materials, that is, the process $p_{iz}$ must collect all the required raw materials before it can be processed once, and this is recorded as a processing batch. When $p = 1$, it is the first sub-process of each level process. The raw materials of this sub-process may depend on a variety of semi-finished products of the previous level process where $i = i' + 1$; when $p > 1$, it means that it depends on the same level. The semi-finished products of the same product line in the process are used as raw materials, and, at this time, $i = i'$, $p = p' + 1$, and $z = z'$. Table 2 shows the definition of time decision variables:

### 2.2. Model Assumptions

#### 2.2.1. Processing Equipment and Processing Assumptions

- There is only one product per piece of processing equipment, considering that some machines process different proportions of raw materials to produce various products. This article defines them as separate machines, i.e., each machine is set to handle one type of product. In an actual environment, the parameters of a machine are different when producing different products. During the production process of an assembly line, the machine does not automatically adjust settings.
- The processing equipment responsible for the same process has the same processing performance.

**Table 1.** Symbol definitions. AGV: automated guided vehicle.

| Symbol | Meaning |
| --- | --- |
| $I$ | Process level |
| $i$ | Number of process levels |
| $Z_i$ | Number of product types output by the *ith* operation to the next operation |
| $z_i$ | The $z$ product on the *ith* process; when $i = 0$, it means this is the starting process |
| $N_{z_I}$ | Final product type $z$ quantity |
| $P_{iz}$ | The number of operations owned by the $z$ product line of the ith operation |
| $p_{iz}$ | The p operation of the *ith* level operation of $z$ product line |
| $R^{+}_{p_{iz}p'_{i'z'}}$ | The number of products from the operation $p'_{i'z'}$ required to process the operation of a batch $p_{iz}$ |
| $R^{-}_{p_{iz}}$ | Number of products produced in one processing batch of the process $p_{iz}$ |
| $L_{p_{iz}}$ | A total batch of operations $p_{iz}$ |
| $l_{p_{iz}}$ | The *Ith* of the operation $p_{iz}$ |
| $J^{+}_{p_{iz}p'_{i'z'}}$ | The total number of products $p_{iz}$ required for the operation $p'_{i'z'}$ |
| $j^{+}_{p_{iz}p'_{i'z'}}$ | The process $p_{iz}$ comes from the *jth* raw material of the process $p'_{i'z'}$. When $p = 1$, it is represented as the entrance process of each stage of the process. At this time, there are a variety of products that depend on the previous process. Where $i = i' + 1$ and when $p > 1$, it means that the process is relying on the products in the same process as that of the raw materials; at this time, $i = i'$, $p = p' + 1$, and $z = z'$ |
| $J^{-}_{p_{iz}}$ | Total number of finished products processed by the operation $p_{iz}$ |
| $j^{-}_{p_{iz}}$ | The *jth* finished product of the process $p_{iz}$ |
| $K_{p_{iz}}$ | Total number of execution machines $p_{iz}$ operations |
| $k_{p_{iz}}$ | Machine $k$ in process $p_{iz}$ |
| $V$ | Total AGV |
| $t_{p_{iz}}$ | The time required for a machine on the $p_{iz}$ to process unit raw materials |

**Table 2.** Variable definitions.

| Decision Variables | Meaning |
| --- | --- |
| $S_{p_{iz}lk}$ | The *Ith* batch of the operation $p_{iz}$ starts on the machine $k_{p_{iz}}$ |
| $T_{p_{iz}lk}$ | The end time of the *Ith* batch of operation $p_{iz}$ is processed on the machine $k_{p_{iz}}$ |
| $T_{jp_{iz}k}$ | Processing completion time $j^{-}_{p_{iz}}$ on $k_{p_{iz}}$ |
| $C_{jp_{iz}p'_{i'z'}}$ | Raw materials $j^{+}_{p_{iz}p'_{i'z'}}$ shipping to the $p_{iz}$ of the process |
| $t_{p_{iz}p'_{i'z'}}$ | AGV transit time between operation $p_{iz}$ and operation $p'_{i'z'}$ |
| $S'_{C_{jp_{iz}p'_{i'z'},v}}$ | Car $v$ starts $C_{jp_{iz}p'_{i'z'}}$, the transportation task of the without-load phase |
| $T'_{C_{jp_{iz}p'_{i'z'},v}}$ | Car $v$ starts $C_{jp_{iz}p'_{i'z'}}$) the end time of the without-load phase of the transportation mission |
| $S_{C_{jp_{iz}p'_{i'z'},v}}$ | Car $v$ starts $C_{jp_{iz}p'_{i'z'}}$ the start time of the load phase of the transportation mission |
| $T_{C_{jp_{iz}p'_{i'z'},v}}$ | Car $v$ starts $C_{jp_{iz}p'_{i'z'}}$ the end time of the load phase of the transportation mission |
| $x_{C_{jp_{iz}p'_{i'z'},v}}$ | $= \begin{cases} 1, & C_{jp_{iz}p'_{i'z'}} \text{ By car } v \text{ transport} \\ 0, & \text{other} \end{cases}$ |
| $y_{klp_{iz}}$ | $= \begin{cases} 1, & \text{*Ith* batch on operation } p_{it} \text{ is processed on a machine } k_{p_{iz}} \\ 0, & \text{other} \end{cases}$ |

### 2.2.2. AGV Transportation Route Assumptions

1. The transportation efficiency of each AGV cart is the same, and the speed remains the same during transportation [33].
2. The capacity of each AGV can only transport unit quantities of raw materials, and the mass and volume of each unit of raw materials does not exceed the AGVs rated load.
3. There is sufficient avoidance space at intersections and around the equipment, and the AGV avoidance passage time is negligible.
4. AGV loading and unloading time are included in the transportation time.
5. The vehicle has no faults during transportation [34,35].

6. The model only considers the process that requires AGV transportation and simplifies the process that does not require AGV transportation. The first-stage process is the first-stage process that requires an AGV to participate in shipping, and the last-stage process is what happens subsequently.

*2.3. Object Function*

In this study, the AGV scheduling plan with the highest production efficiency in the drawing shop was studied. The objective function was to minimize the time required for the drawing shop to complete all finished products, i.e., to reduce the maximum completion time. Here, Equation (1) defines U as the complete set.

$$T_{max} = \min \left\{ \max_{jp_{iz} \in \cup, k \in [1, k_{P_{Iz}}]} T_{iP_{Iz}k} \right\} \tag{1}$$

*2.4. Uniqueness Constraint*

Unique constraints include those on processing materials, production equipment, and the AGV.

2.4.1. Raw-Material Processing Uniqueness

Each processing raw material corresponds one-to-one with the process, batch, and production equipment. In this case, only the goods to be transported by the AGV cart can be determined to determine the transport route of the cart and a set that restricts one batch under one process on one machine. Equation (2) defines the input raw materials of a specified process that can only be attributed to one batch under the procedure for processing. Equation (3) represents the constraint on the batch to which raw materials belong in the first step of each stage of the production line. Equation (4) indicates that the batch belongs to the raw material constraint in other levels of each stage of the production line. The products of a specified process can only be produced by a batch under this process, as shown in Equation (5).

$$\sum_{k=1}^{k_{P_{iz}}} y_{klp_{iz}} = 1 \tag{2}$$

$$\forall i \in [1, I],\ z \in [1, Z], p \in [1, P_{iz}], l \in \left[1, L_{P_{iz}}\right]$$

$$\sum_{l=1}^{L_{P_{iz}}} u_{jlp_{iz}p'_{i'z'}} = 1 \tag{3}$$

$$p = 1,\ p' = P_{i'z'} = i - 1,\ \forall i \in [1, I],\ z \in [1, Z],\ z' \in [1, Z_{i'}]$$

$$\sum_{l=1}^{L_{P_{iz}}} u_{jlp_{iz}p'_{i'z'}} = 1 \tag{4}$$

$$p > 1,\ p' = P_{iz} - 1, i' = i, z' = z,\ \forall i \in [1, I],\ z \in [1, Z],\ p \in [1, P_{iz}]$$

$$\sum_{l=1}^{L_{P_{iz}}} w_{jlp_{iz}} = 1 \tag{5}$$

$$\forall i \in [1, I], z \in [1, Z_i], p \in [1, P_{iz}], j \in \left[1, J_{p_{iz}}\right]$$

2.4.2. AGV Uniqueness Constraint

Raw material j is transported only in one AGV transportation process, as shown in Equations (6) and (7). Since the first process of each stage of the production line is different from other means, the situation of $p$ is discussed separately in Equations (6) and (7).

$$\sum_{v=1}^{V} x_{C_{jP_{iz}P'i'z'}v} = 1 \tag{6}$$

$$P = 1, \; P' = P_{i'z'}, \; i' = i - 1$$

$$\forall i \in [1, I], z \in [1, Z_i], z' \in [1, Z_{i'}], j \in \left[1, J_{p_{iz}p'i'z'}\right]$$

$$\sum_{v=1}^{V} x_{C_{jP_{iz}P'i'z'}v} = 1 \tag{7}$$

$$P > 1, \; P' = P_{iz} - 1, \; i' = i, z' = z$$

$$\forall i \in [1, I], z \in [1, Z_i], p \in [1, P_{iz}], j \in \left[1, J_{p_{iz}p'i'z'}\right]$$

### 2.4.3. Unique Constraints of Production Equipment

Concerning the limitation on processing equipment, a machine processes only one raw material at a time, and this constraint is determined by the attributes of the processing equipment for different processes in the drawing shop, as shown in Equations (8) and (9). When $p = 1$, the raw materials for machining may come from different production lines. When $p > 1$, the raw materials for machining come from the same production line. Additionally, one machine can only produce one product at a time. Equation (10) defines the processing equipment of the same process that provides the same number of products in one operation, and the number of products produced by the processing equipment of different methods is not necessarily the same.

$$\sum_{j=1}^{J^+ P_{iz}P'i'z'} u_{jlp_{iz}p'i'z'} = R^+_{p_{iz}p'i'z'} \tag{8}$$

$$p = 1, p' = P_{i'z'}, i' = i - 1$$

$$\forall i \in [1, I], z \in [1, Z_i], z' \in [1, Z_{i'}], l \in \left[1, L_{p_{iz}}\right]$$

$$\sum_{j=1}^{J^+ P_{iz}P'i'z'} u_{jlp_{iz}p'i'z'} = R^+_{p_{iz}p'i'z'} \tag{9}$$

$$p > 1, p' = P_{iz} - 1, i' = i, z' = z$$

$$\forall i \in [1, I], z \in [1, Z_i], p \in [1, P_{iz}], l \in \left[1, L_{p_{iz}}\right]$$

$$\sum_{j=i}^{J^- p_{iz}} w_{jlp_{iz}} = R^-_{p_{iz}} \tag{10}$$

$$\forall i \in [1, I], z \in [1, Z_i], p \in [1, P_{iz}], l \in \left[1, L_{p_{iz}}\right]$$

## 3. Multi-AGV Scheduling Simulation Modeling

The case-study used here was mainly based on a multi-AGV scheduling mathematical model and an optimal batch distribution strategy. Based on the Jingwei Wuxi ring spinning production process, Siemens Plant Simulation was used to establish a multi-AGV scheduling simulation model for the drawing shop. The scheduling strategy built a simulation scenario and analyzed the impact of different AGV numbers on scheduling performance, i.e., completion time and some factors that affect the configuration of AGV numbers.

*3.1. Production Operation Drawing Workshop Simulation Input*

By exploiting state-of-the-art mechanical technology, we considered the production of two product types:

The first was the 55%/45% product, which comprises 55% cotton and 45% polyester in a mixed process. The second was the 60%/40% product, which comprised 60% cotton and 40% polyester in a mixed process. Table 3 shows the cotton production process parameters, and Table 4 shows the polyester production process parameters. The process parameters of product one (55%/45%), are shown in Table 5. The process parameters of product two (60%/40%) are shown in Table 6. Table 7 shows the raw material requirements for each device (regardless of the binocular device). Table 8 shows the processing time of each device.

In our simulation, each region initialized an AGV resource pool. Depending on the set policy, the regions where the AGV could operate were different. AGVs ran differently between different areas. Table 9 shows the connectivity and running time between different areas. The drawing shop layout was divided into four areas, and the first area was pre-combined for combing, lap-forming, and combine combing; this was called 'Region A.'

**Table 3.** Cotton processing parameters.

| Equipment | Quantity | Speed (m/min) | Tampons Specification (m) |
| --- | --- | --- | --- |
| Pre-drawing combs | 2 | 400 | 800 |
| Drawing Roller | 1 | 120 | 10 |
| Combs | 5 | 350 | 700 |

**Table 4.** Polyester processing parameters.

| Equipment | Quantity | Speed (m/min) | Tampon Spec. in Cotton Barrels (m) |
| --- | --- | --- | --- |
| Polyester strip | 2 | 288 | 560 |

**Table 5.** 55%/45% product process parameters.

| Equipment | Quantity | Speed (m/min) | Thread Specification (m) |
| --- | --- | --- | --- |
| First Mixing | 1 | 380 | 650 |
| Second Mixing | 1 | 380 | 750 |
| Third Mixing | 1 | 380 | 850 |

**Table 6.** 60%/40% product process parameters.

| Equipment | Quantity | Speed (m/min) | Thread Specification (m) |
| --- | --- | --- | --- |
| First Mixing | 1 | 380 | 4000 |
| Second Mixing | 1 | 380 | 4500 |
| Third Mixing | 1 | 380 | 5000 |

**Table 7.** Equipment processing batch information.

| Equipment | Quantity of Raw Materials Required | A Single Batch OF Semi-Finished Products/Total |
| --- | --- | --- |
| Polyester strip | 5 Barrels | 1 barrel/5 in total |
| Pre-drawing combs | 5 Barrels | 1 barrel/5 in total |
| Drawing roller | 24 Barrels | One lap/30 in total |
| Combing Frame | 8 Laps | 1 barrel/5 in total |
| 65%/45% First Mixing | C (Cotton) 3 and T (polyester) 3 | 1 barrel/6 in total |
| 60%/40% First Mixing | C (Cotton) 4 and T (polyester) 3 | 1 barrel/7 in total |
| Second Mixing | 6 Barrels | 1 barrel/6 in total |
| Third Mixing | 6 Barrels | 1 barrel/6 in total |

**Table 8.** Equipment processing time.

| Equipment | Single Batch Processing Time (Min) | Single Product Processing Time |
| --- | --- | --- |
| Polyester strip | 10 | 2 min |
| Pre-drawing combs | 10 | 2 min |
| Drawing roller | 2 | 40 s |
| Combs | 16 | 2 min |
| 55%/45% First Mixing | 10 | 100 s |
| 60%/40% First Mixing | 10 | 100 s |
| Second Mixing | 15 | 150 s |
| Third Mixing | 15 | 150 s |

**Table 9.** AGV running time between different areas.

| Starting Area | Final Area | Operation Hours/S |
| --- | --- | --- |
| Area 1 (A) | Area 2 (C) | 10 |
| Area 2 (A) | Area3 (B) | 12.5 |
| Area 3 (B) | Area4 (D) | 10 |

The production efficiency of the production line was analyzed under different AGV quantities throughout the multiple simulations [36]. The number of AGVs in the resource pool of each area increased from 1 to 3, and a total of 81 groups of simulations were performed. AGV1, AGV2, AGV3, and AGV4 signify the number of AGV initializations for the four regions. Completion time refers to the time required to process all products, and the production requirements of the two products were set up to 50. The following is a comparative analysis of the impact of the number of AGV allocation factors and its effect on the completion time in the application scenario of the batch distribution strategy.

*3.2. Simulation Model Construction*

3.2.1. Machine Module

The machine tool of the draw frame was characterized by the need to process multiple raw materials at a time. Various finished products were produced one after another, so the simulation could understand that the input and output were different in one processing batch. To describe this type of machine in 'Siemens Tecnomatix Plant Simulation,' multiple necessary components were required to implement this feature. The composition of the combed pre-parallel device, which divided into two major modules, is schematically shown in Figure 4. The first module mainly realized the functionality that a processing batch needed a fixed amount of raw materials, and the second module realized the processing of the processed materials after the required raw materials were satisfied.

Component assembly was introduced for continuous manufacturing to correctly understand the function of processing various raw materials in one processing batch. A single processor was used to constrain the processing time of a shipment after the number of raw materials was satisfied, upon which the machine started processing, executed the 'Method' method, and released the exit of 'Buffer3.' Then, the entities in 'Buffer3' entered 'Buffer1.' When there was an entity flowing out of 'Buffer3,' the 'Method1' method was executed to count the number of products that could be produced by a processing batch. Afterward, the entities in 'Buffer1' flowed out one after another. The constraint requirement that a processing batch needed to process multiple raw materials at the same time in order to successively produce several products was realized.

The simulation implementation of a machine that required multiple raw materials was similar, except that, in this case, there were numerous input ports. Taking a device with a mixed process as an example, Figure 5 shows the mixed-process machine composition of a 55%/45% product.

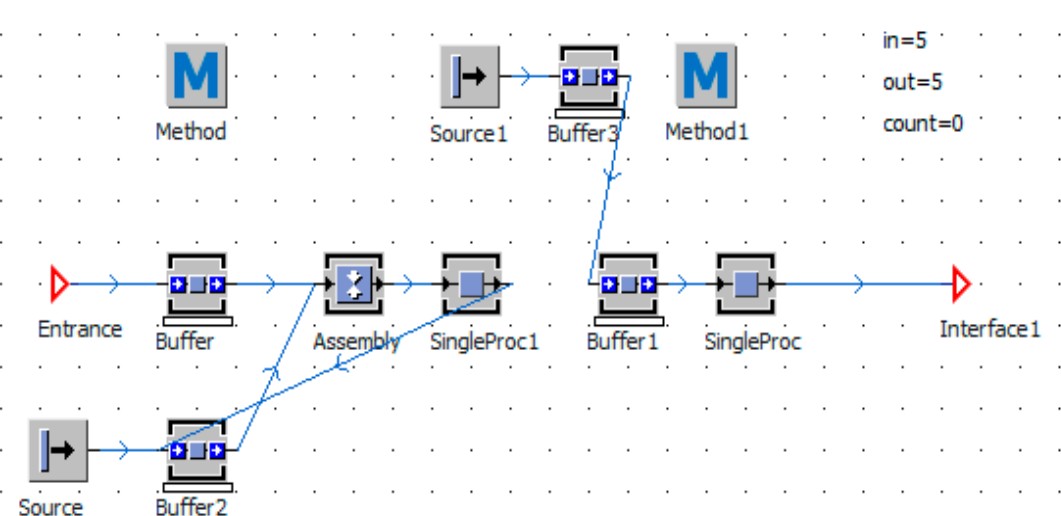

**Figure 4.** Automatic assembly system simulation module in plant simulation.

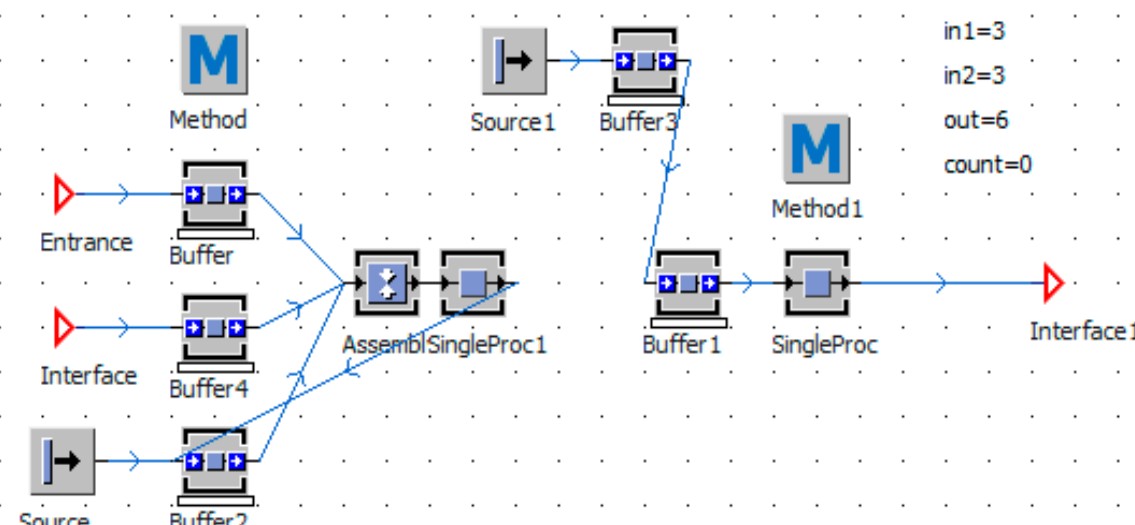

**Figure 5.** Mixed-process machine simulation composition diagram.

### 3.2.2. AVG Transportation Simulation

The "workers" in the resource were used to simulate AGV transportation. In the simulation model building, the function of the AGV was to transport the product from one process to another process, and this could be achieved by "workers." Figure 6 shows the loading point of one process, the driving route of the AGV, and the unloading point of the next process. The single processor at the entrance was the unloading point of the AGV, which needed to set the exiting property of the processor. Then, the AGV could carry the goods to the specified location and then unload; as such, the AGV was simulated in this way.

### 3.2.3. Scheduling Policy

The machine selection strategy of the tight node was controlled by a flow-controller, where the plan set in Exit Strategy was that of the cyclic-sequence time, and the corresponding list of the configured batch distribution policy could complete the setting. When the cyclic time was selected, the rotation distribution was given; this distribution laid evenly, as shown in Figure 7.

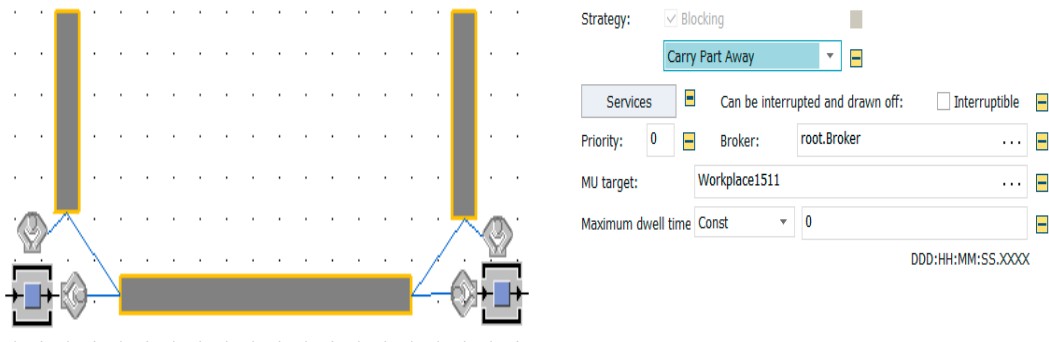

**Figure 6.** AGV loader simulation.

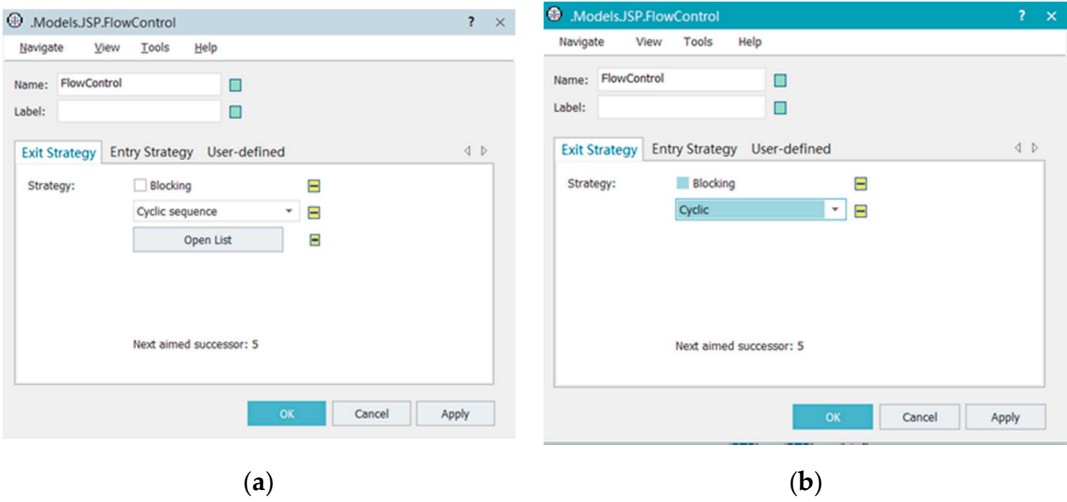

(**a**)            (**b**)

**Figure 7.** (**a**,**b**) show the evenly distributed batch simulation implementation.

In plant-simulation resource pool policy, setting up the same broker for the machining machine for the same regional operation allowed for the operation within the region to share the AGV. The broker, which was the part of the 'Tecnomatix Plant Simulation environment,' mostly cooperated with the exporter and the importers of the station, the parallel station, the assembly station, and the dismantling station. If it was a cross-regional independent resource pool policy, different brokers were configured for different neighborhoods. AGVs in the various areas could not call each other; implementing a cross-zone shared resource pool set brokers for multiple regions as the same as those for the connecting regions.

### 3.2.4. Simulation Model

Figure 8 shows the extensive simulation model of the multi-AGV flow-shop scheduling workshop.

Based on the simulation scene of the shop floor assembly line, the following analysis simulation experiment was carried out.

First was the AGV cross-regional shared resource pool and the AGV cross-regional independent resource pool policy comparison. The second was under the optimal production equipment distribution strategy and AGV resource pool strategy. The influence of AGV quantity on multi-AGV scheduling performance and the analysis of AGV quantity allocation factors was also considered. Through the above analysis, the results of multi-AGV scheduling decision-making were obtained. Because the machine selection strategy was based on which batch distribution was better than the uniform distribution strategy, the cross-regional independent resource pool strategy and cross-regional shared resource pool strategy had an inconsistent relationship in different situations. From there, we went on

to combine the actual case of the spinning workshop and to further compare the performance of the two resource pool strategies.

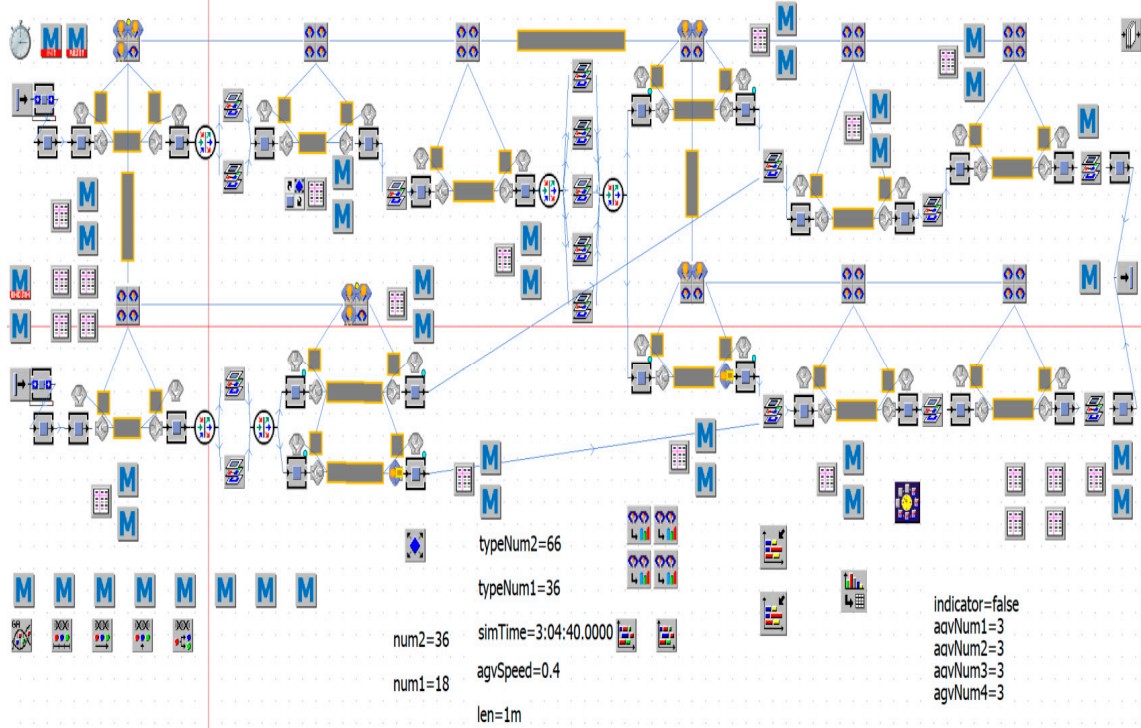

**Figure 8.** Execution of the simulation.

### 3.3. AGV Resource Pool Strategy Based on Bottleneck Analysis

In order to improve simulation efficiency, a production efficiency analysis was carried out on each production process. Since the critical issue in this paper was the production process involving AGV transportation, the procedure before raw material cotton and polyester was not considered, nor was the third mixing process. The cotton production was a serial assembly line process. Therefore, the output efficiency of the raw cotton material was related to the bottleneck process in these three mixing processes. The raw material consumption and finished output speed of the three processes of cotton raw materials were as follows.

1.  Pre-drawing combs: Each machine produced five cans every 10 min with a total of 2 machines.
2.  Drawing roller: Each machine consumed 24 cans every 2 min and produced 30 cotton rolls, with a total of 1 machine.
3.  Combs: Each machine consumed eight cotton bobbins every 16 min and produced five cotton laps with a total of 5 machines;

In the polyester production process, there was only a polyester blending process. The efficiency of the blending process was such that each machine produced five cans per 10 min with a total of 2 machines. Regardless of the transportation time, the output efficiency of the cotton raw materials was equal to the output efficiency of the bottleneck process in the three processes, assuming that the output of the immediately preceding process in each of the above processes was sufficient. The process of pre-drawing combs produced an average of one cotton can per minute. The consumption of 12 cotton bobbins yielded 15 cotton laps, while the process of combing consumed 40 cotton bobbins every 16 min. The bottleneck process was pre-drawing combing, and the final cotton raw material output efficiency was an average of 5 cotton rolls produced every 4 min. Because there was only one process in polyester,

the output efficiency of the polyester was the output efficiency of the process. An average of one polyester cylinder was produced per minute. Figure 9 shows the production process.

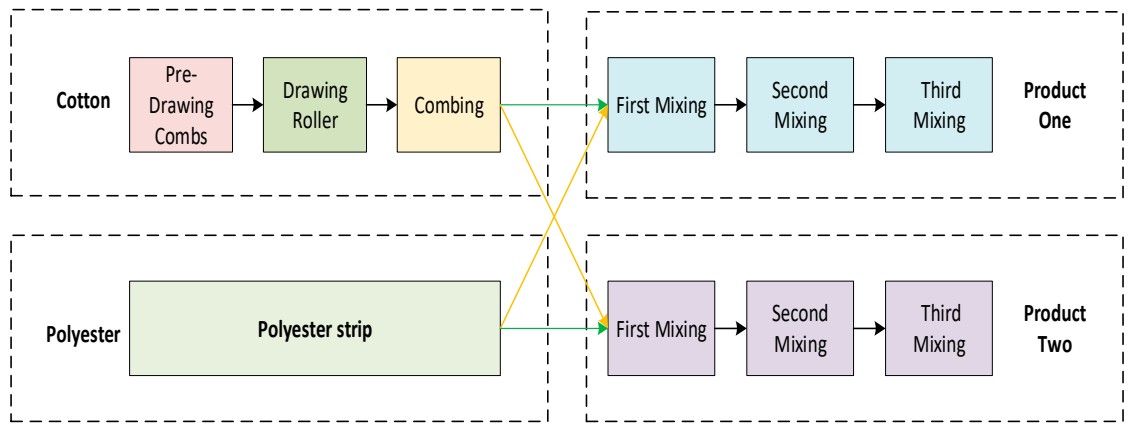

**Figure 9.** Process flow diagram.

In the mixed process of cotton and polyester, different machines could be used for different products, and the required ratio of raw materials was also different. However, the same processes existed for different products. However, machines were not shared between various products, so the subsequent process was still a continual process for each product. The bottleneck process determined the output. The following were the inputs and outputs of the first, second, and third mixing processes for the two products.

### 3.3.1. Product One (55%/45%)

Each machine consumed three cotton rolls and three polyester drums every 10 min in its first mixing process and produced a total of six cans. In the second and third mixing process with one machine, it consumed six cans in every 15 min to produce six cans.

### 3.3.2. Product Two (60%/40%)

Each machine consumed four cotton rolls and three polyester cans every 10 min in its first mixing process, yielding a total of seven cans. In the second and third mixing process with one machine, it consumed six cans in every 15 min to produce six cans.

If the supply of raw materials was sufficient, the efficiency of product one consumed three cotton rolls and three polyester cans every 15 min, thus yielding six cans. Moreover, the efficiency of product two consumed four cotton rolls and three polyester cans every 15 min, thus yielding seven cans. The second and third mixing processes were the bottleneck processes. Based on the raw material processing and mixing process in the spinning and drawing workshop, every 16 min, the raw material processing process provided raw material at an average speed of 25 cotton cans and 16 polyester cans. In the mixing process, when all the machines were turned on, the efficiency was such that it consumed seven cotton cans and six polyester cans every 15 min on average, thus producing 6 of 'product one,' 6 of 'product 2,' and a total of 12 products. According to the efficiency comparison ratio in the material section and the mixing process section, it was found that the bottleneck was in the mixing process section. The output efficiency of the cotton pre-drawing combs and polyester strips was the same as the bottleneck of raw material. Finally, the process of mixing one, two, and three became the output bottleneck.

According to the workshop scheduling theory, the bottleneck process is the crucial section that restricts the output of an assembly line. According to the above analysis, the bottleneck process was identified. Therefore, if the output speed of the assembly line needed to increase, the output capacity of the bottleneck process needed to be increased as well. In the above calculations, the time required

for the product to move between the processes was ignored. When we considered the handling time for the goods to transport, the output was further reduced. Still, due to the existence of the bottleneck process, the machine utilization of other non-bottleneck processes was not full. In practical applications, if transportation resources are sufficient and the transportation time is shorter than the product processing time, then the transportation time could be ignored. However, after the introduction of an AGV, transportation resources become scarce. Different transportation tasks compete for limited transportation resources. Therefore, the next section focuses on analyzing how to schedule capacity resources to meet the processing requirements of each machine and to produce a specified number of products in the shortest time.

## 4. Simulation Results Analysis

### 4.1. Comparative Analysis of Two AGV Resource Pool Strategies

4.1.1. Analysis of Simulation Results Based on Cross-Regional Shared Resource Pools

As shown in Table 10, for the simulation result under the cross-regional shared resource pool, according to the number of AGVs in four regions, the number was in the form of "A1-B1-C1-D1," indicating that regions A, B, C and D each had an AGV. The following is a principal analysis of the changes in the number of AGVs between different regions at the time of completion. Under the shared resource pool across regions, different AGV quantities affected the completion time (as shown in Table 10).

Figure 10 shows a line chart of the maximum completion time under the different number of AGVs, where the horizontal axis is the number of AGVs and the vertical axis is the completion time. With the increase in the number of AGVs, the completion time gradually decreased until the number of AGVs was 10. The completion time was stable, and adding AGVs did not significantly improve production efficiency. The process takes time to produce was the main reason for this, and in terms of process, the time it took to produce one batch per as was longer. Thus, increasing the number of AGVs could significantly improve productivity when AGV resources are scarce. Nevertheless, as the number of AGVs increased, the production efficiency of the process gradually became a new bottleneck, and it was not possible to further improve the production efficiency by merely increasing the number of AGVs.

From Table 10, another problem can be seen, i.e., different allocations still affected the completion time at the same AGV. The following are examples of processes with six and nine AGVs, and the effect of various AGV allocation schemes on the completion time is analyzed. In the programs mentioned above, the total number of AGVs for a total of 10 distribution programs was six, the total number of AGVs in 16 programs was nine, and Figure 11 shows the completion time for each application with a whole a production time of 6 h. The largest allocation of completion time was that of "A1-B1-C1-D3;" the smallest was that of "A2-B2-C1-D1," which showed that under the cross-regional shared resource pool, region A and C could connect, B and D could be combined, and A and B were connected. Under "A2-B2-C1-D1," it could be understood that region A and C had three AGVs, and B and D had three AGVs; the allocation resources was more uniform than the number of AGVs in region A, except in "A1-B1-C1-D3." The completion time difference between the remaining programs was not significant when the number of AGVs in region A was 2; there was a clear downward trend between the three schemes, where "A2-B1-C1-D2" was higher than "A2-B1-C2-D1," and A2-B2-C1-D1" was the smallest. The difference between "A2-B1-C2-D1" and "A2-B2-C1-D1" was that area D scheduling took too long, resulting in the consumption of too much time on empty-load operations, while A2-B1-C2-D1 tilted resources to the raw material area, i.e., for cotton and polyester production and processing. This indicated that in the current production environment, raw material production time accounted for the entire production time, which was longer and needed to tilt towards more capacity resources. "A2-B2-C1-D1" had the lowest completion time because of its uniform distribution of resources, which was flexible to dispatch.

**Table 10.** Shared resource pool simulation results across regions.

| Number of AGV in Regions | Complete Working Hours | Serial Number | Complete Working Hours | Serial Number | Complete Working Hours |
|---|---|---|---|---|---|
| A1-B1-C1-D1 | 11:31:22 | A2-B1-C1-D1 | 9:22:47 | A3-B1-C1-D1 | 8:19:21 |
| A1-B1-C1-D2 | 10:32:43 | A2-B1-C1-D2 | 8:50:37 | A3-B1-C1-D2 | 7:58:13 |
| A1-B1-C1-D3 | 9:25:48 | A2-B1-C1-D3 | 7:40:32 | A3-B1-C1-D3 | 7:02:43 |
| A1-B1-C2-D1 | 10:01:01 | A2-B1-C2-D1 | 8:14:34 | A3-B1-C2-D1 | 7:00:07 |
| A1-B1-C2-D2 | 8:35:47 | A2-B1-C2-D2 | 7:48:03 | A3-B1-C2-D2 | 7:20:05 |
| A1-B1-C2-D3 | 7:37:24 | A2-B1-C2-D3 | 7:01:26 | A3-B1-C2-D3 | 6:39:55 |
| A1-B1-C3-D1 | 8:43:39 | A2-B1-C3-D1 | 7:39:03 | A3-B1-C3-D1 | 6:29:23 |
| A1-B1-C3-D2 | 8:19:02 | A2-B1-C3-D2 | 6:54:04 | A3-B1-C3-D2 | 6:06:10 |
| A1-B1-C3-D3 | 6:58:07 | A2-B1-C3-D3 | 6:15:16 | A3-B1-C3-D3 | 5:53:56 |
| A1-B2-C1-D1 | 8:43:14 | A2-B2-C1-D1 | 7:47:08 | A3-B2-C1-D1 | 7:24:25 |
| A1-B2-C1-D2 | 8:37:37 | A2-B2-C1-D2 | 8:24:59 | A3-B2-C1-D2 | 7:43:04 |
| A1-B2-C1-D3 | 7:39:17 | A2-B2-C1-D3 | 7:17:48 | A3-B2-C1-D3 | 6:30:45 |
| A1-B2-C2-D1 | 8:39:55 | A2-B2-C2-D1 | 7:36:28 | A3-B2-C2-D1 | 7:02:18 |
| A1-B2-C2-D2 | 8:27:33 | A2-B2-C2-D2 | 7:00:09 | A3-B2-C2-D2 | 6:18:14 |
| A1-B2-C2-D3 | 7:18:25 | A2-B2-C2-D3 | 6:31:08 | A3-B2-C2-D3 | 5:43:51 |
| A1-B2-C3-D1 | 7:18:21 | A2-B2-C3-D1 | 6:12:34 | A3-B2-C3-D1 | 5:52:14 |
| A1-B2-C3-D2 | 7:02:31 | A2-B2-C3-D2 | 6:16:59 | A3-B2-C3-D2 | 6:01:39 |
| A1-B2-C3-D3 | 6:22:01 | A2-B2-C3-D3 | 5:55:06 | A3-B2-C3-D3 | 5:33:38 |
| A1-B3-C1-D1 | 8:11:15 | A2-B3-C1-D1 | 7:08:49 | A3-B3-C1-D1 | 6:55:14 |
| A1-B3-C1-D2 | 7:55:27 | A2-B3-C1-D2 | 7:02:44 | A3-B3-C1-D2 | 6:17:15 |
| A1-B3-C1-D3 | 6:47:07 | A2-B3-C1-D3 | 6:11:53 | A3-B3-C1-D3 | 5:32:05 |
| A1-B3-C2-D1 | 7:22:26 | A2-B3-C2-D1 | 7:28:21 | A3-B3-C2-D1 | 6:08:00 |
| A1-B3-C2-D2 | 6:52:03 | A2-B3-C2-D2 | 7:05:10 | A3-B3-C2-D2 | 5:49:54 |
| A1-B3-C2-D3 | 6:13:16 | A2-B3-C2-D3 | 6:05:57 | A3-B3-C2-D3 | 5:42:54 |
| A1-B3-C3-D1 | 6:31:31 | A2-B3-C3-D1 | 6:02:41 | A3-B3-C3-D1 | 6:02:54 |
| A1-B3-C3-D2 | 6:37:05 | A2-B3-C3-D2 | 6:06:14 | A3-B3-C3-D2 | 5:36:09 |
| A1-B3-C3-D3 | 5:51:02 | A2-B3-C3-D3 | 5:39:36 | A3-B3-C3-D3 | 5:25:27 |

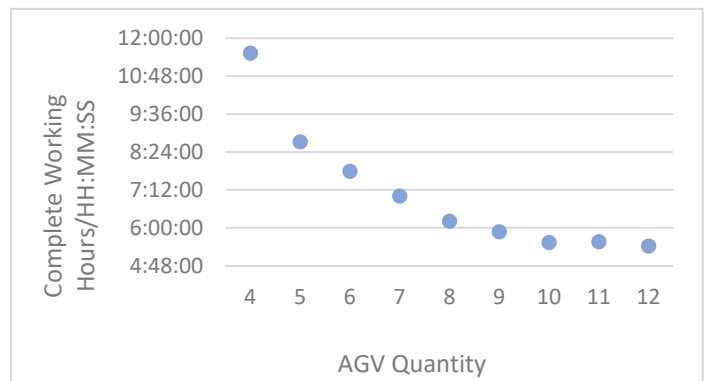

**Figure 10.** Quantity and completion time diagram.

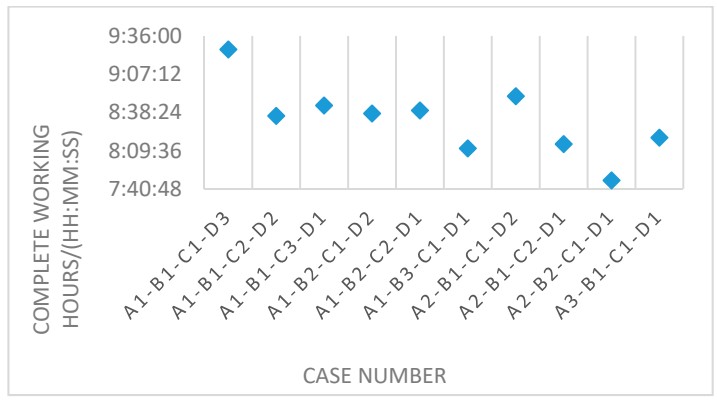

**Figure 11.** Completion time under 6 AGVs.

Figure 12 presents a comparison of the completion times under nine AGVs, with "A2-B3-C2-D2" having the longest completion time in the 16 distribution schemes, while "A3-B2-C3-D1" had the shortest completion time. In the 16 scenarios, the completion time varied little, with it being the same when 6 AGVs were present and the raw material stage required more resource tilt.

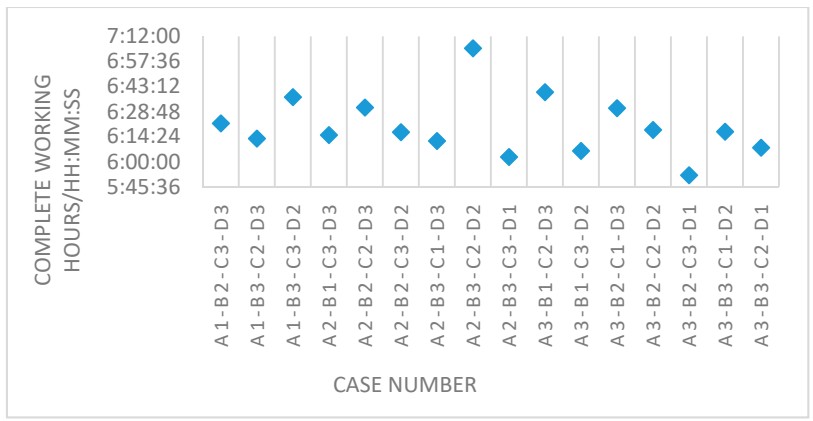

**Figure 12.** Completion time under 9 AGVs

In addition to the maximum completion time, the utilization rate of AGVs needed to be analyzed. Figure 13 is the AGV utilization rate under "A3-B3-C3-D3." From the Figure, we can see that the utilization rate of an AGV was not high. The time of AGV empty operation in regions A and B was relatively high, and so the utilization rate was relatively high as well; meanwhile, the time of AGV empty operation of areas C and D was low. The overall utilization rate of each AGV was only 10–20%. In comparison, the utilization rate of AGV of region D was less than 10%, indicating that the number of AGVs was over-allocated and resources overflowed.

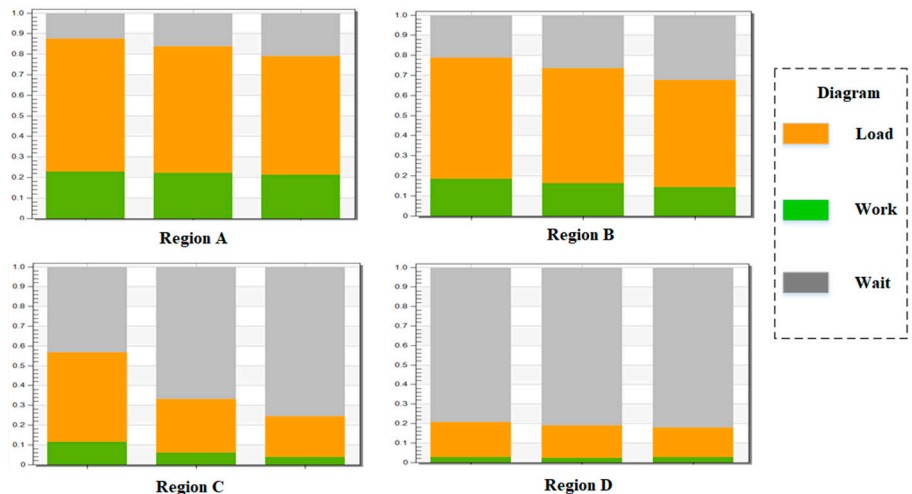

**Figure 13.** Utilization comparison ratio.

4.1.2. Analysis of Cross-Regional Independent Resource Pool Simulation Results

Table 11 shows the result of the simulation under a distinct pool of resources across regions. As with the cross-regional shared resource pool policy, the effect of changes in the number of AGVs in the different areas on the time of completion was analyzed.

**Table 11.** Cross-regional independent resource pool simulation results.

| Number of AGV in Regions | Complete Working Hours | Serial Number | Complete Working Hours | Serial Number | Complete Working Hours |
|---|---|---|---|---|---|
| A1-B1-C1-D1 | 6:03:47 | A2-B1-C1-D1 | 4:45:26 | A3-B1-C1-D1 | 4:46:56 |
| A1-B1-C1-D2 | 4:43:11 | A2-B1-C1-D2 | 3:26:48 | A3-B1-C1-D2 | 3:52:52 |
| A1-B1-C1-D3 | 4:39:46 | A2-B1-C1-D3 | 3:07:50 | A3-B1-C1-D3 | 3:33:54 |
| A1-B1-C2-D1 | 4:51:03 | A2-B1-C2-D1 | 4:26:58 | A3-B1-C2-D1 | 4:34:54 |
| A1-B1-C2-D2 | 4:43:11 | A2-B1-C2-D2 | 3:27:01 | A3-B1-C2-D2 | 3:48:21 |
| A1-B1-C2-D3 | 4:39:46 | A2-B1-C2-D3 | 3:03:25 | A3-B1-C2-D3 | 3:32:55 |
| A1-B1-C3-D1 | 5:16:27 | A2-B1-C3-D1 | 4:15:58 | A3-B1-C3-D1 | 4:48:31 |
| A1-B1-C3-D2 | 4:49:32 | A2-B1-C3-D2 | 3:24:12 | A3-B1-C3-D2 | 3:51:17 |
| A1-B1-C3-D3 | 4:39:46 | A2-B1-C3-D3 | 3:02:48 | A3-B1-C3-D3 | 3:32:15 |
| A1-B2-C1-D1 | 5:09:27 | A2-B2-C1-D1 | 4:38:58 | A3-B2-C1-D1 | 4:32:39 |
| A1-B2-C1-D2 | 5:01:03 | A2-B2-C1-D2 | 3:26:47 | A3-B2-C1-D2 | 3:17:17 |
| A1-B2-C1-D3 | 4:56:37 | A2-B2-C1-D3 | 3:02:40 | A3-B2-C1-D3 | 2:54:20 |
| A1-B2-C2-D1 | 5:17:42 | A2-B2-C2-D1 | 4:38:56 | A3-B2-C2-D1 | 4:32:35 |
| A1-B2-C2-D2 | 5:02:16 | A2-B2-C2-D2 | 3:26:44 | A3-B2-C2-D2 | 3:15:14 |
| A1-B2-C2-D3 | 4:56:37 | A2-B2-C2-D3 | 3:02:40 | A3-B2-C2-D3 | 2:49:25 |
| A1-B2-C3-D1 | 5:17:42 | A2-B2-C3-D1 | 4:25:16 | A3-B2-C3-D1 | 4:07:00 |
| A1-B2-C3-D2 | 5:02:16 | A2-B2-C3-D2 | 3:23:19 | A3-B2-C3-D2 | 3:08:05 |
| A1-B2-C3-D3 | 4:56:37 | A2-B2-C3-D3 | 3:02:40 | A3-B2-C3-D3 | 2:48:18 |
| A1-B3-C1-D1 | 5:12:50 | A2-B3-C1-D1 | 4:39:08 | A3-B3-C1-D1 | 4:23:51 |
| A1-B3-C1-D2 | 5:01:03 | A2-B3-C1-D2 | 3:26:47 | A3-B3-C1-D2 | 3:17:17 |
| A1-B3-C1-D3 | 4:56:37 | A2-B3-C1-D3 | 3:02:40 | A3-B3-C1-D3 | 2:54:20 |
| A1-B3-C2-D1 | 5:17:42 | A2-B3-C2-D1 | 4:38:56 | A3-B3-C2-D1 | 4:23:35 |
| A1-B3-C2-D2 | 5:02:16 | A2-B3-C2-D2 | 3:26:44 | A3-B3-C2-D2 | 3:12:59 |
| A1-B3-C2-D3 | 4:56:37 | A2-B3-C2-D3 | 3:02:40 | A3-B3-C2-D3 | 2:49:25 |
| A1-B3-C3-D1 | 5:17:42 | A2-B3-C3-D1 | 4:20:08 | A3-B3-C3-D1 | 4:07:00 |
| A1-B3-C3-D2 | 5:02:16 | A2-B3-C3-D2 | 3:22:02 | A3-B3-C3-D2 | 3:08:05 |
| A1-B3-C3-D3 | 4:56:37 | A2-B3-C3-D3 | 3:02:40 | A3-B3-C3-D3 | 2:48:18 |

Figure 14 shows a line chart of the maximum completion time under different AGV numbers under the cross-region independent resource pool strategy. The horizontal axis is the number of AGVs, and the vertical axis is the completion time. When the number of AGVs increased, the completion time gradually decreased. Until the number of AGVs was 10, the completion time tended to be stable, as in the cross-region shared resource pool. The reason for this is that the completion time under the cross-region independent resource pool was much shorter than the completion time under the cross-region shared resource pool. When ten units were configured with AGVs, the completion time was only about 2 h and 50 min.

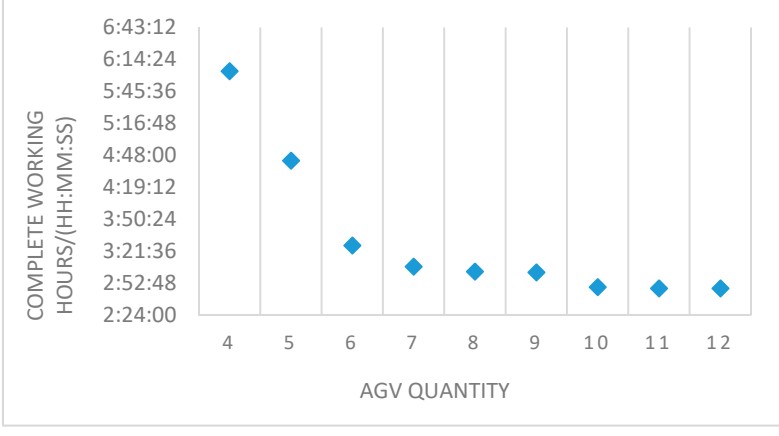

**Figure 14.** AGV quantity and completion time map.

Increasing the number of AGVs did not significantly improve production efficiency for the same reasons as sharing resource pools across the regions. Moreover, the minimum completion time of the same ratio under the two resource allocation strategies was compared, and it was found that the cross-region shared resource pool required close to 5 h, and the cross-region independent resource pool had a minimum completion time of 2 h and 48 min, which was far less than the cross-region shared resource pool. The five hours required for the cross-region shared resource pool indicated that in the current simulation configuration, independent resource pools across regions could effectively improve production efficiency.

As well as sharing resource pools across regions, we analyzed the simulation results of the same number of AGVs under different configuration schemes. Here, for example, the number of AGVs was 7 and 10, and the effects of various AGV allocation schemes on the completion time were analyzed. In the above scenarios, the total number of AGVs for the 16 allocation programs was seven. There were 10 scenarios with a complete a number of AGVs of 10. Figure 15 shows the completion time for each program with total damage of 7 h. The largest allocation of completion time was for "A1-B3-C2-D1," and the smallest was for "A2-B1-C1-D3."

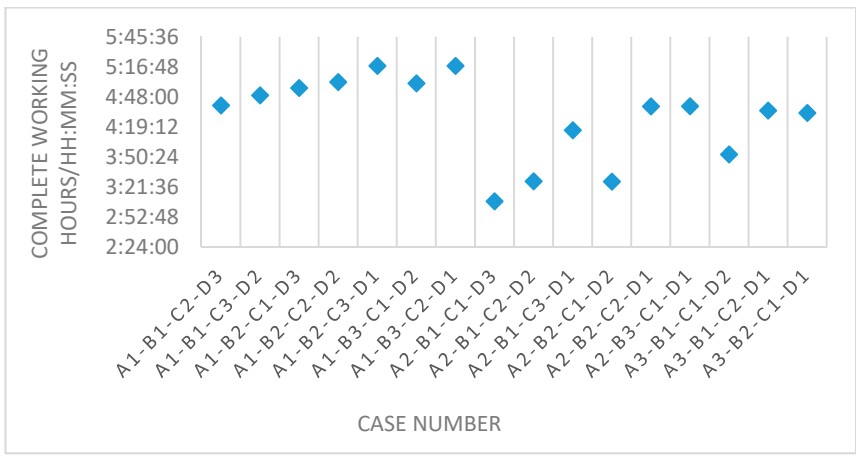

**Figure 15.** Completion time under 7 AGVs.

Figure 16 shows a comparison of the completion times under 10 AGVs, for which A1-B3-C3-D3 had the longest completion time in the 10 distribution schemes, whereas "A3-B2-C2-D3" had the least completion time. In all ten scenarios, the completion time varied a little, same as each scenario with the 7 AGVs, and the raw material stage required more resource tilt.

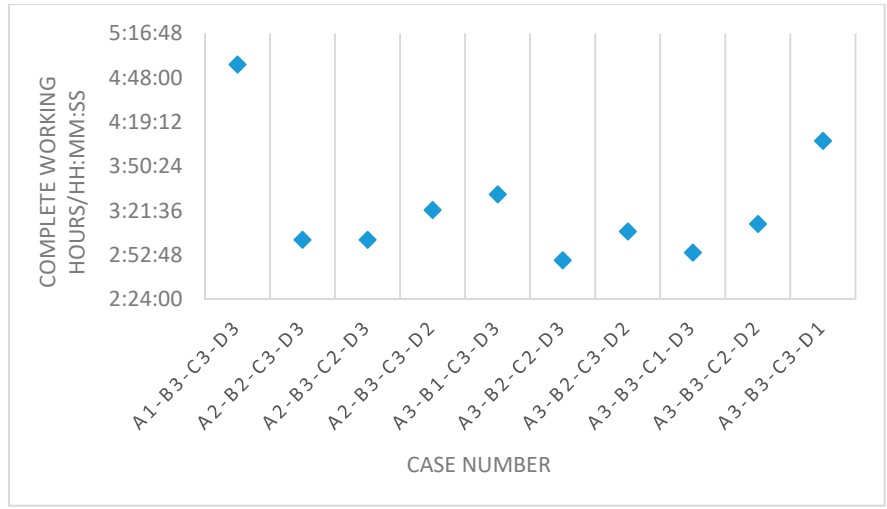

**Figure 16.** Completion time under 10 AGVs.

Unlike shared resource pools across regions, the overall utilization of AGVs was higher because AGV cross-regional scheduling was not allowed under the cross-regional independent resource pool policy. Figure 17 presents a comparative analysis of AGV utilization under a separate resource pool across regions, with regions A and C corresponding to cotton and polyester raw material areas, respectively, while AGVs were waiting most of the time in regions B and D, indicating that AGVs were over-resourced.

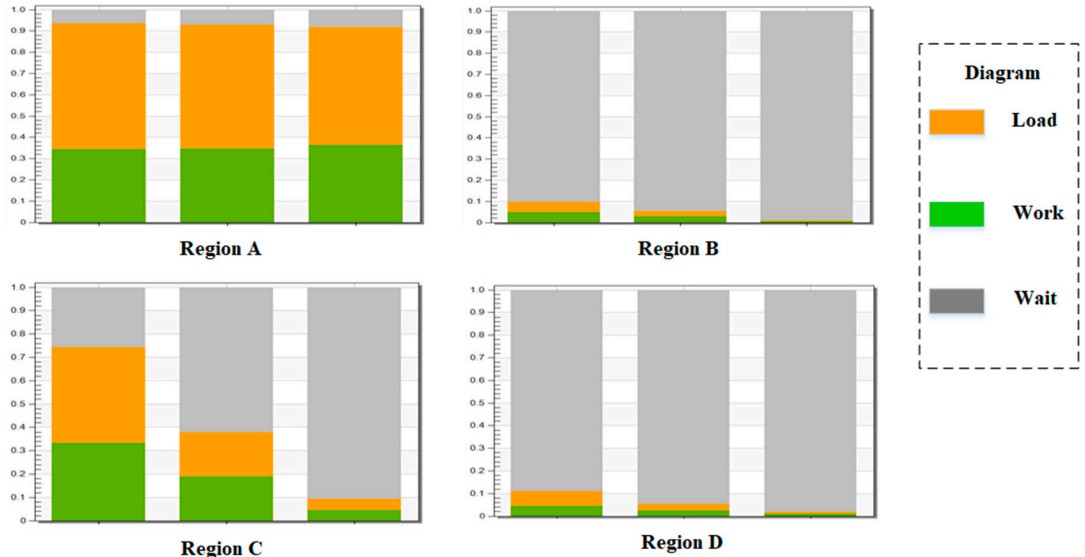

**Figure 17.** AGV Utilization comparison.

Based on the above comparison analysis (which originates in practice), due to the high load caused by cross-regional scheduling, the cross-regional independent resource pool was superior to the cross-regional shared resource pool in an actual situation, whether in the case of an AGV resource shortage or when the AGV resources are sufficient. With other unchanged parameters, the distance between regions will affect the time of cross-region scheduling. When the distance is more significant, cross-regional scheduling increases the load rate, which affects the completion time. When the distance is small, cross-regional scheduling can effectively improve the utilization of AGV resources and shorten completion time.

### 4.2. Comprehensive Analysis of Multi-AGV Scheduling in the Workshop

To further analyze the characteristics of the multi-AGV scheduling problem in the drawing shop, the number of required products was increased to analyze whether the AGV scheduling resources would still tilt toward the raw material area.

#### 4.2.1. Increase the Number of Products

Figure 18 shows the scheduling time when the number of two products was 100, and Figure 19 shows scheduling time when the number of two products was 200. The results of the analysis presented as complete working hours and the number of experiments according to their confidence interval. It can be observed that between Experiment 1 and Experiment 3, the average value of the completion time significantly declined, and it showed a periodic law. There was a regularity between the completion time and the AGV configuration. The reason for this is that the cycle amount when increasing the number of AGVs was 3, which means that increasing the number of AGVs could significantly improve production efficiency and shorten completion time. Additionally, this proved that when the number of products doubled, the AGV resource was insufficient, and the completion time nearly doubled, as well. However, when the AGV resources were relatively sufficient, the completion time did not

increase proportionally, indicating that the production line was not in a balanced state and the number of products produced was very small. Not all the machines were working for a long time during production, and there was a substantial warm-up period. The effect of increasing the number of AGVs on the completion time was not very obvious, but when the number of products increased, the warm-up period gradually decreased during the process. In conclusion, increasing the number of AGVs could significantly affect completion time until the AGV resources are saturated.

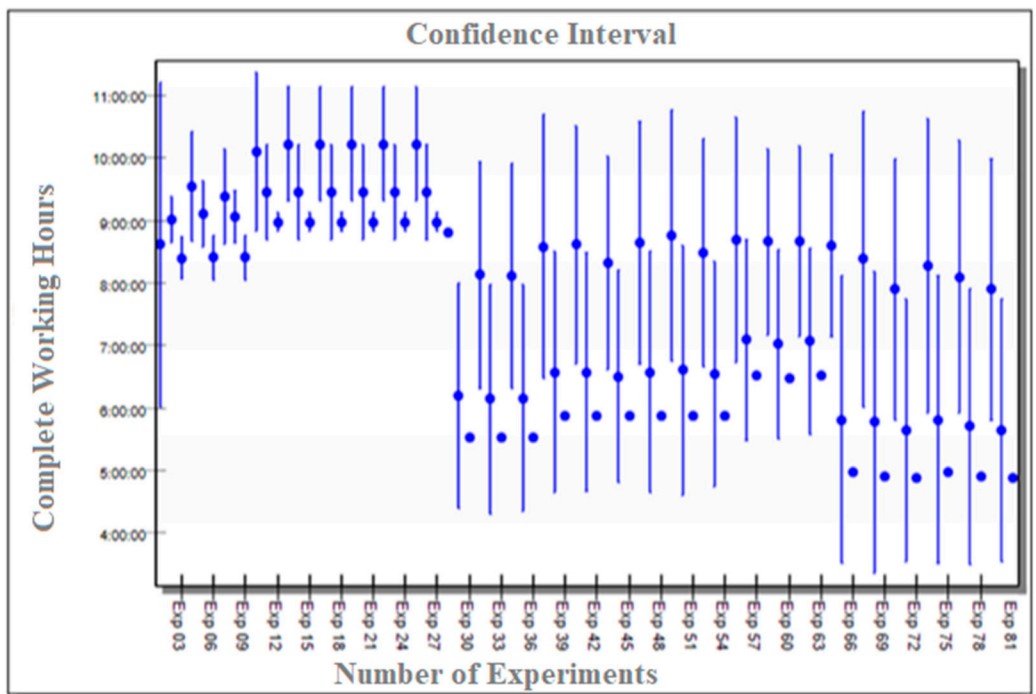

**Figure 18.** Completion time of 100 h under the individual resource pool.

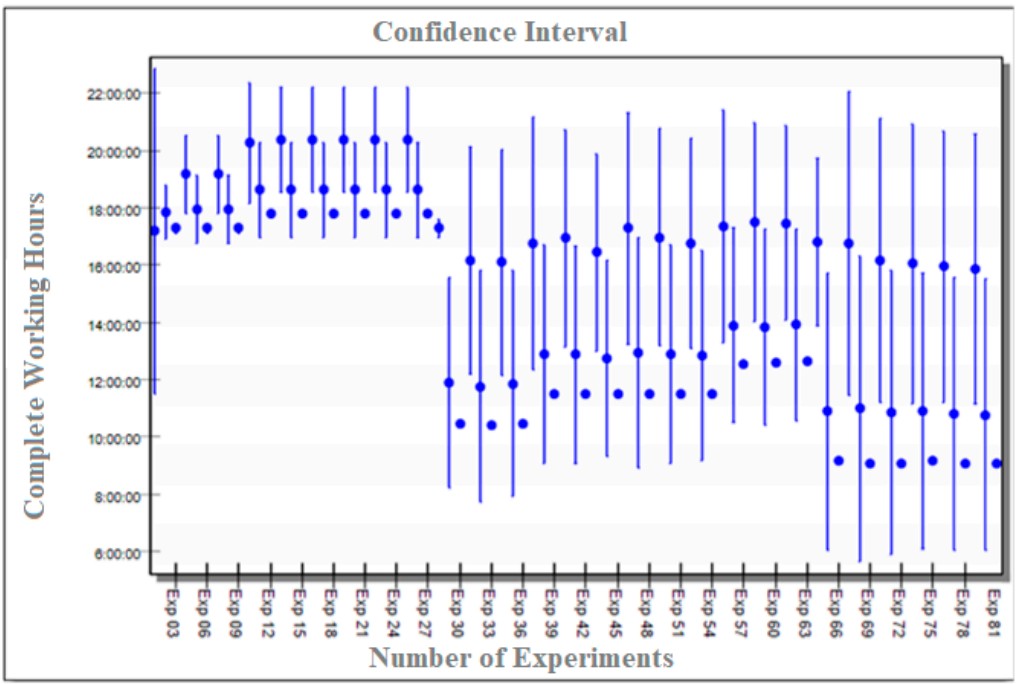

**Figure 19.** The number of products under the independent resource pool was 200 h of completion time each.

### 4.2.2. AGV Quantity Decision Analysis

The factors that need to be considered in the decision of the number of AGVs are the completion time and the utilization rate of each AGV. Here, we analyzed a set of data regarding the unit time contribution of each AGV, i.e., the increase in the contribution of an AGV to the completion time. Under the cross-region shared resource pool strategy, increasing the number of AGVs could effectively shorten the completion time, but under different circumstances, increasing the reduction rate of AGVs had different performances. Figure 20 shows that when the number of AGVs increased from four to five, the impact on the completion time was massive, and the completion time reduction rate was close to 25%, before it gradually flattened. After the number of AGVs reached nine, the impact was small—completion time stabilized. Figure 21 shows the impact of the increase in AGV numbers on the completion time under the strategy of independent resource pools across regions. The completion time had a more significant impact at six AGVs, and when eight AGVs were used, the completion time stabilized.

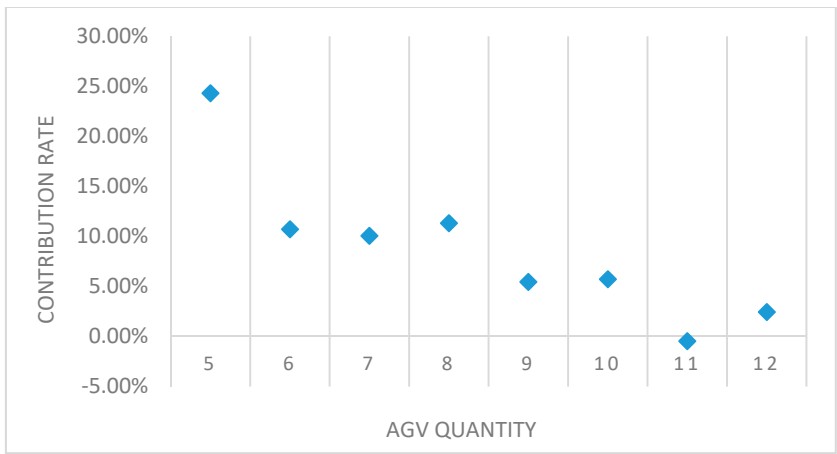

**Figure 20.** Impact of AGV quantity on completion time under the cross-regional shared resource pool.

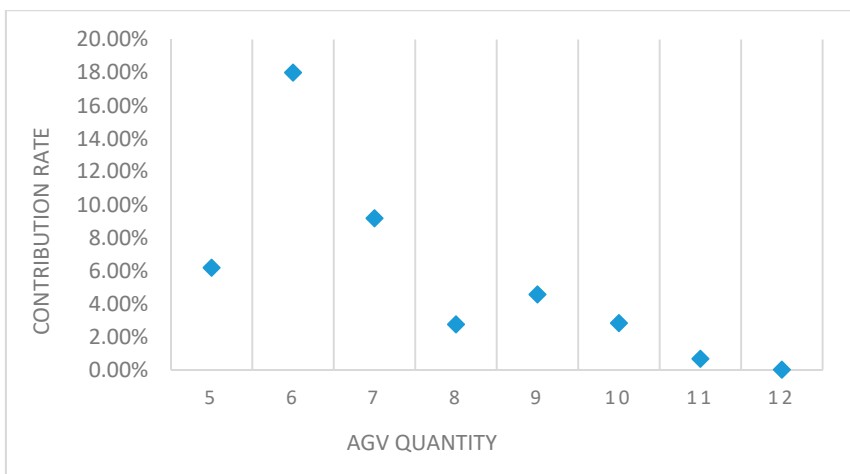

**Figure 21.** Impact of AGV quantity on completion time under a cross-regional independent resource pool.

According to the number of AGVs based on the completion time and the above analysis of increasing the number of products, we could estimate that when the number of products was small, a relatively small number of AGVs could be selected to increase the utilization of AGVs. When the number of products was significant, we needed to increase the number of AGVs to shorten the completion time.

### 4.2.3. Optimization Analysis

Figure 22 shows multiple AGV scheduling scenarios by the Gantt chart using manual scheduling. By comparing the completion time, it could be found that, whether it was a cross-region shared resource pool or a cross-region independent resource pool, the completion time was less than the completion time of the manual scheduling job, which illustrates the effectiveness of the method. The introduction of the AGV transport of semi-finished products in a drawing shop can reduce the labor intensity of workers, improve transport efficiency, and shorten completion time.

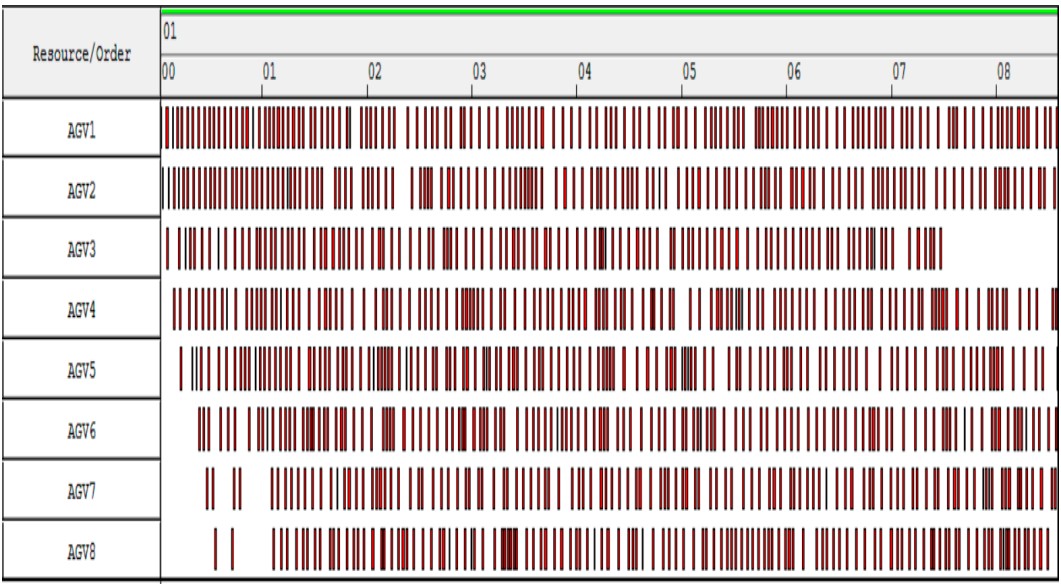

**Figure 22.** Manual dispatch job Gantt diagram.

### 5. Conclusions and Future Work

In this paper, we investigated the actual production process parameters and characteristics of highly distributed manufacturing system like textile ring-spinning combing section. This work was inspired to resolve the confronting problems of scheduling, real-time can distribution, and path planning in the continuous production of spinning by benefitting the mixed flow-shop predictive modelling. To effectively reduce the makespans and total completion time, it was vital to define the properties and features of the workshop, equipment, products, and AGVs. Based on the two AGVs scheduling strategies, a novel approach of handling both cross-regional shared resource and cross-regional independent resource pools was analyzed. For the dissimilar cotton and polyester draw-out processing, we established an overall mathematical model of multi-AGV scheduling to solve the problems of can distributions and to prevent conflict and deadlock by assigning different tasks: AGV assignment, AGV sorting, and task source. Moreover, for the intended categories of scheduling tasks, an AGV transportation route strategy was also developed for mass production in a spinning CPPS. The extensive computational experiments were performed using 'Siemens Tecnomatix Plant Simulation software,' according to the production of a certain number of products and two scheduling strategies. The simulation results analysis of 81 groups of optimization targets with completion time showed a specific range of AGVs. As the number of AGVs increased, the completion time decreased. The number of AGVs reached a certain threshold, and the completion time stabilized. On this basis, the utilization rate and the completion time of the products were also analyzed. When the number of AGVs rose to a certain extent, the contribution of increasing AGV numbers to the completion time decreased sharply, thus reducing the utilization rate of AGVs. By comparing the results of 81 sets of simulations under the two strategies, it was found that the cross-regional independent resource pool strategy was better than the cross-regional shared resource pool strategy under the actual scenario.

These results demonstrated the adequacy of the methods we used and proved that flow-shop predictive modeling for when multi-AGV resources are scarce also produces, for each AGV, a control mode and, if necessary, a preventive maintenance plan. Based on the comparison with our scheduling approach, it was found that the results of multi-AGV scheduling have distinct advantages that can significantly shorten completion time.

In the future, it will be attractive to examine multi-AGV scheduling by applying different scheduling algorithms in order to investigate the potential mechanisms responsible for transportation task set scheduling decisions using a genetic algorithm.

**Author Contributions:** B.F. and Q.M. Wen mainly conceptualized the idea for the study and were responsible for project administration. Q.M. Wen was also responsible for preprocessing all the data and for the formal analysis by investigating the accuracy of the results. B.F. wrote the initial draft of the manuscript and was responsible for investigating resources, revising, and improving the manuscript according to the reviewer's comments. All the work was done under the supervision and guidance of J.B. All authors have read and agreed to the published version of the manuscript.

**Funding:** This research was funded by the National key research and development program of China (Grant No. 2017YFB1304000).

**Acknowledgments:** This work would not have been achievable without the support of the head of 'Institute of Intelligent Manufacturing' Donghua University. The authors also appreciate the anonymous referee's for their valuable and profound comments, which have improved the quality of the paper.

**Conflicts of Interest:** The authors declare no conflict of interest.

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
