# Peer review of "Flow-Shop Predictive Modeling for Multi-Automated Guided Vehicles Scheduling in Smart Spinning Cyber–Physical Production Systems"

_electronics, doi:10.3390/electronics9050799_

Round 1

Reviewer 1 Report

  1. What is the novelty of the paper?
  2. The author wrote "smart spinning" in the title of the paper, but the author did not explain the meaning of the smart spinning in the paper.
  3. It is better the author writes Multi-Automated Guided Vehicle than Multi-AGV in the title of the paper.
  4. The author wrote flow-shop predictive modeling in the title but the author wrote "predictive control of AGV in abstract. (Line 21). It is better the author explains the flow-shop predictive modeling in the abstract. I think the author is only explain the predictive control of AGV. I can't find the flow shop predictive modeling for multi-AGV.
  5. Equation 1, 2, 3, 4, 5, 6, 7, 8, 9, and 10. If the author cited from the papers, please write the number of the papers in references the author cited for those equations.
  6. The author wrote table 4 in line 204. It is better the author writes it in the top of the table.
  7. In table 6 (line 208), the author wrote the units of the single batch processing time in (m). I think that time is minute or second.
  8. In the page 12 (line 322-350), The line spacing is bigger than the other lines.
  9. In conclusions (line 513), the author wrote " We established an overall mathematical model of multi-AGV Scheduling". Please show the mathematical model of multi-AGV Scheduling.
  10. Figure 11, 12, 15 and 16, the graphs are drawn too sharp. It is better those graphs are drawn smooth.

Author Response

Point 1: What is the novelty of the paper?

Response 1:  This is the first time a novel approach handling both cross-region shared resource pool, and inter-regional independent resource pool have been presented within the context of ‘ring-spinning’ in the textile industry specifically for combing. For this, we established an overall mathematical model of multi-AGV scheduling to solve the problems of can distributions and to prevent conflict and deadlock by assigning it different tasks; AGV assignment, AGV sorting, and task source. Flow-shop predictive modeling for multi-AGV resources are scarce; it also produces, for each AGV, the control mode and, if essential, the preventive maintenance plan.

Point 2: The author wrote “smart spinning” in the title of the paper, but the author did not explain the meaning of the smart spinning in the paper.

Response 2: Thank you for pointing out this. The text explaining smart spinning is now included for a better understanding.

“As discussed in [2], a smart spinning system is a system that contains multiple computing elements (sensors, actuators) and a processing unit, all managed by restraining the data. Based on the flow of information, data is pre-processed, and a command is sent to the actuator to perform a pre-programmed action that can detect changes in their surroundings and react to them to produce a practical outcome. [From line 41 to 45]

Point 3: It is better the author writes Multi-Automated Guided Vehicle than Multi-AGV in the title of the paper.

Response 3: Thank you for your valuable suggestion. It is changed as suggested.

“Flow-Shop Predictive Modeling for Multi-Automated Guided Vehicles Scheduling in Smart Spinning Cyber-Physical Production Systems”

Point 4: The author wrote flow-shop predictive modeling in the title, but the author wrote, “Predictive control of AGV in the abstract. (Line 21). It is better the author explains the flow-shop predictive modeling in the abstract. I think the author only explains the predictive control of AGV. I cannot find the flow shop predictive modeling for multi-AGV.

Response 4: Thanks for picking this up. The changes have been made accordingly, and it can be seen in the abstract [Line 15, 19, 23, 24 and 25].

Point 5: Equation 1, 2, 3, 4, 5, 6, 7, 8, 9, and 10. If the author cited from the papers, please write the number of the papers in references the author cited for those equations.

Response 5: We apologize for the confusion. All the equations have been modified and corrected. Furthermore comprehensive mathematical model [From Line 135 to Line 182] has been added for better understanding.

Point 6: The author wrote table 4, inline 204. It is better the author writes it at the top of the table.

Response 6: Thanks for pointing this. It is adjusted as requested.

Point 7: In table 6 (line 208), the author wrote the units of the single batch processing time in (m). I think that time is minute or second.

Response 7: Thank you for your comment. It is corrected as pointed out, and the single batch processing time unit is indeed in minutes (min).

Point 8: On page 12 (line 322-350), the line spacing is bigger than the other lines.

Response 8: Thanks for picking this up. It is adjusted as requested.

Point 9: In conclusions (line 513), the author wrote: “We established an overall mathematical model of multi-AGV Scheduling.” Please show the mathematical model of multi-AGV scheduling.

Response 9: Thank you for raising this point. The detailed “mathematical description” has been added in section 2.1. Mathematical model of multi-AGV scheduling, 2.1.1. Processing equipment definition, 2.1.2. Raw material and product definition, 2.1.3. AGV definition, and 2.1.4. Overall variable definition. [From Line 135 to Line 182]

Point 10: Figures 11, 12, 15, and 16, the graphs are drawn too sharp. It is better those graphs are drawn smoothly.

Response 10: Thank you very much for discovering this. The graphs in Figures 11, 12, 15, and 16 have been modified according to the suggestions.

Reviewer 2 Report

Topic:

The paper addresses the issue of AGV scheduling and path planning in order to avoid conflicts and deallocks during production process, using the example of textile factory. The topic is interesting and refers to the problems of modern manufacturing.

Compliance with the journal scope:

The topic of the paper math the journal scope in the area of multiagent systems, however it should be better emphasized in the content.

Language style:

The paper is written in a manner that makes it hard to read. The content needs thorough revision, because of grammar errors, typos and missing words.

References:

The references are correclty selected and are up-to-date. All of the references are cited in the content.

Introduction and research motivation:

The topic of research is clearly stated, however it should be better referred to the short literature review, done in the "Introduction" section. It is not only about expected benefits, but also about answering the question why the elaborated methods are more advanced than the presented ones.

The methods and scientific background:

  • The presentation of the developed methods must be improved.
    • there is no detailed desciption of the methods, used in the research,
    • it is not indicated if the presented math formulas are elaborated by Authors or are cited from other sources (in this case it is obligatory to cite the source),
    • the symbols in formulas are (in majority) not explained,
    • the regions, defined in the paper, should be explained and shown in a illustration,
    • the term "Broker" (lines 264, 266) needs more explanation,
    • the conclusions (section 5) are too general and sometimes obvious, moreover there is no connection with the goal, presented in lines 71-77: what about conflicts and deadlocks? what about the novelty (comparison between new approach and the existing ones)?

Other comments:

  • mathematical formulas are hard to read - I don't know if it is processing or formatting issue,
  • try to avoid single-paragraph sections,
  • explain 55/45 and 60/40 terms before use - now the explanation is in the line 199 while first occurence is in the line 195,
  • why the columns in the Tables 8 and 9 are named "Serial numbers" not (for example) "Number of AGV in regions"?
  • please do not use lines to connect points in charts, where the X axis has discrete values, like "AGV Quantity" or "Case number",
  • Fig. 18, Fig. 19 - what is the "Exp" value on X axis?
  • lines 59-63 - the wrong names of authors were used (Ralf, Marcin, Waldemar are probably the first names)

Author Response

The paper addresses the issue of AGV scheduling and path planning in order to avoid conflicts and deadlocks during the production process, using the example of a textile factory. The topic is interesting and refers to the problems of modern manufacturing.

  1. Compliance with the journal scope:

Point 1: The topic of the paper matches the journal scope in the area of multiagent systems. However, it should be better emphasized in the content.

Response: Thanks for your valuable comment. The other reviewer also suggested, so the title of the manuscript have been modified accordingly.

“Flow-Shop Predictive Modeling for Multi-Automated Guided Vehicles Scheduling in Smart Spinning Cyber-Physical Production Systems”

  1. Language style:

Point 2: The paper is written in a manner that makes it hard to read. The content needs thorough revision because of grammar errors, typos, and missing words.

Response: Thank you so much for your suggestions and for picking this up. The structure and language have been reorganized to improve the logic of the paper. The whole paper has been proof-read to ensure proper use of Grammar and vocabulary without any errors, typos, or missing words.

  1. References:

Point 3: The references are correctly selected and are up-to-date. All of the references are cited in the content.

Response: Thank you very much for your appreciation.

  1. Introduction and research motivation:

Point 4: The topic of research is clearly stated. However, it should be better referred to as the short literature review done in the “Introduction” section. It is not only about expected benefits, but also about answering the question of why the elaborated methods are more advanced than the presented ones.

Response: Thank you for your suggestion. The introduction part has been modified according to the suggestion, and the literature review thoroughly reviewed by adding new content to describe the elaborated method in a better way. For details Line 41-45, 62, 63, 83-85, and 89, can be seen.

  1. The methods and scientific background:

Point 5: The presentation of the developed methods must be improved. There is no detailed description of the methods used in the research. It is not indicated if the presented math formulas are elaborated by Authors or are cited from other sources (in this case it is obligatory to cite the source), the symbols in formulas are (in the majority) not explained, the regions, defined in the paper, should be explained and shown in an illustration, the term “Broker” (lines 264, 266) needs more explanation, the conclusions (section 5) are too general and sometimes obvious. Moreover, there is no connection with the goal, presented in lines 71-77: what about conflicts and deadlocks? What about the novelty (comparison between a new approach and the existing ones)?

Response: We apologize for the confusion. All the formulas have been modified and corrected. Furthermore comprehensive mathematical model [From Line 135 to Line 182] is added for better understanding. The detailed “mathematical description” can be seen on section 2.1. Mathematical model of multi-AGV scheduling, 2.1.1. Processing equipment definition, 2.1.2. Raw material and product definition, 2.1.3. AGV definition, and 2.1.4. Overall variable definition

Broker, as mentioned in [lines 264, 266] explained clearly for better understanding.

Conclusions thoroughly reviewed and modified according to the reviewer’s suggestion, and some text further added within the context of the goal. The changes can be seen on [Lines 595-602, 607,608, 610-613].

If we talk about the “Novelty,” This is the first time a novel approach handling both cross-region shared resource pool, and inter-regional independent resource pool have been presented within the context of ‘ring-spinning’ in the textile industry specifically for combing. For this, we established an overall mathematical model of multi-AGV scheduling to solve the problems of can distributions and to prevent conflict and deadlock by assigning it different tasks; AGV assignment, AGV sorting, and task source. Flow-shop predictive modeling for multi-AGV resources are scarce; it also produces, for each AGV, the control mode and, if essential, the preventive maintenance plan.

  1. Other comments:
  2. Mathematical formulas are hard to read - I do not know if it is processing or formatting issue, try to avoid single-paragraph sections,

Response: Thanks for pointing out this. It has been formatted again to avoid the difficulty in reading.

  1. Explain 55/45 and 60/40 terms before use - now the explanation is in line 199 while the first occurrence is in line 195,

Response:  Thank you for picking this up. It has been modified [Line 277 and 278] according to the suggestions. 

  • Why the columns in Tables 8 and 9 are named “Serial numbers” not (for example) “Number of AGV in regions”?

Response: Thank you so much for noticing this, it has been changed accordingly.

  1. Please do not use lines to connect points in charts, where the X-axis has discrete values, like “AGV Quantity” or “Case number,”

Response: Thanks for pointing out and suggestions. Figures 10, 11, 12, 14, 15, and 16 modified as suggested.

  1. Fig. 18, Fig. 19 - what is the “Exp” value on X-axis?

Response: Thank you for picking this. It has been added as pointed.

  1. Lines 59-63 - the wrong names of authors were used (Ralf, Marcin, Waldemar are probably the first names)

Response: We apologize for the mistake. It is corrected as suggested.

Reviewer 3 Report

Overall, I believe that this manuscript is well written and organized and represents a good contribution. However, only few corrections need to be addressed:

  • The variables and indices in equations (1) through (10) are not explained. It’s difficult if not impossible for the reader to comprehend. Every new notation must be explained immediately after its first use.
  • Table 4 should be renamed “60/40 Product process parameters”.

Author Response

Overall, I believe that this manuscript is well written and organized and represents a good contribution. However, only a few corrections need to be addressed:

Point 1: The variables and indices in equations (1) through (10) are not explained. It is difficult, if not impossible, for the reader to comprehend. Every new notation must be explained immediately after its first use.

Response: Thanks for highlighting. We apologize for the confusion. All the formulas have been modified and corrected. Furthermore comprehensive mathematical model [From Line 135 to Line 182] is added for better understanding. The detailed “mathematical description” can be seen in section 2.1. Mathematical model of multi-AGV scheduling, 2.1.1. Processing equipment definition, 2.1.2. Raw material and product definition, 2.1.3. AGV definition, and 2.1.4. Overall variable definition

Point 2: Table 4 should be renamed “60/40 Product process parameters.”

Response: Thank you so much for pointing out this mistake. It is renamed as requested. [Line 276]

Round 2

Reviewer 1 Report

1.Please write the equations number correctly

2. Please write the tables correctly

Author Response

Point 1: Please write the equations number correctly.

Response: Thank you so much for the suggestions. We have adjusted the equations accordingly.

Point 2: Please write the tables correctly

Response:  Thanks for pointing out this. From Table 1 to 11, 'Line and Paragraph Spacing, Borders' have been adjusted according to the template. Besides, the font style and size in Table 10 [Line 421] and Table 11[Line 473] have adjusted too.

Reviewer 2 Report

I've read the new version of the paper and found it improved, however I still have some reservations about the Introduction nad Conclusions sections.

Regarding the "Introduction" section, I assume that lines 46-77 presents the literature review, but it is done very generally. I am convinced that there is something that has stimulated the Authors to act, because the development or improvement of a method is always caused by imperfections or limitations of other methods. Therefore, I would expect that the introduction will discuss "competitive" methods in more detail, with attention paid to what inspired the Authors (why it was important to create the new approach?). Also in the "Conclusions" section I would expect the short comment about assets of the new method. Figuratively speaking, mathematics also uses different methods to solve problems, but the method itself is a tool - more or less suitable for a given application. Hence the question: what arguments support the use of the method developed by the Authors in relation to other methods (in this particular case, which is the subject of the paper)?

Other comments:

  • there is still unclear what the Broker is. It should be clear for reader whether it is realated to the Authors' method or it is the part of Technomatix Plant Simulation environment?
  • Fig. 18 and 19 - what the "Exp" stands for (experiment?)? It should be clear for readers.
  • I still have reservations about the presentation of the charts. In my opinion, if we have discrete values on the axis (like number of AGVs), the adjacent points should not be connected with a line. The line suggests the possibility of interpolation, but it is impossible to have e.g. five and half of AGV.
  • I found some more language errors in the text, like "about more" (line 33/34); there are also sentences that could be stylistically improved. Please check the text thoroughly.
  • In my version of paper are some parts that have incorrect formatting:
    • formulas in the part starting from the line 224 to line 268,
    • Figure 8 (format) and the caption "jumped" to the next page,
    • there is no gap between Fig. 1 and the paragraph.

Author Response

I've read the new version of the paper and found it improved. However, I still have some reservations about the Introduction and Conclusions sections.

Point 1: Regarding the "Introduction" section, I assume that lines 46-77 presents the literature review, but it is done very generally. I am convinced that there is something that has stimulated the Authors to act because the development or improvement of a method is always caused by imperfections or limitations of other methods. Therefore, I would expect that the introduction will discuss "competitive" methods in more detail, with attention paid to what inspired the Authors (why it was important to create the new approach?). Also, in the "Conclusions" section, I would expect a short comment about the assets of the new method. Figuratively speaking, mathematics also uses different methods to solve problems, but the method itself is a tool - more or less suitable for a given application. Hence the question: what arguments support the use of the method developed by the Authors in relation to other methods (in this particular case, which is the subject of the paper)?

Response: Thank you so much for the detailed comments and suggestions. We do highly appreciate it to improve the quality of our paper. As advised, the introduction section has been modified thoroughly by including the competitive proposed methods presented in our research work.  A comprehensive literature review has discussed from Line 40 to 80. Line 81 to 98 shows the arguments that support the use of the method we developed. Besides, these comments have been made in the conclusion section from Line 576 to 579, 582-583, and 585 to 587, according to the suggestions.

Other comments:

Point 2: there is still unclear what the Broker is. It should be clear for the reader whether it is related to the Authors' method, or is it part of the Tecnomatix Plant Simulation environment?

Response: Thank you for picking this up. It is part of the Tecnomatix Plant Simulation environment and included in the text after implementing on the suggestions. [Line 325-326]

Point 3: Fig. 18 and 19 - what the "Exp" stands for (experiment?)? It should be clear for readers.

Response: Our apologies for the confusion. Further related text included for the readers as suggested. [Line 527-528]

Point 4: I still have reservations about the presentation of the charts. In my opinion, if we have discrete values on the axis (like the number of AGVs), the adjacent points should not be connected with a line. The Line suggests the possibility of interpolation, but it is impossible to have, e.g., five and half of AGV.

Response: Thank you so much for the detail explanation. It is indeed conducive to correcting the charts. Figures 10, 11, 12, 14, 15, 16, 20, and 21 have modified accordingly. [Line 432, 453, 459, 482, 498, 504, 557, and 559]

Point 5: I found some more language errors in the text, like "about more" (Line 33/34); there are also sentences that could be stylistically improved. Please check the text thoroughly.

Response: Thank you so much for the useful suggestion to enhance the productivity of our paper. The author tried their best to review the text thoroughly and improved the writing qualities, as suggested.

Point 6: In my version of the paper are some parts that have incorrect formatting:

formulas in part starting from line 224 to line 268,

Response: Thanks for noticing this. All formulas have been formatted correctly after getting the suggestions. [After line 220, between lines 232-233, between lines 236-237, and between lines 245-246]

Point 7: Figure 8 (format) and the caption "jumped" to the next page; there is no gap between Fig. 1 and the paragraph.

Response: Tha gap has adjusted according to the requirement.  [Line 335]
